# Development of Dicationic Bisguanidine-Arylfuran Derivatives as Potent Agents against Gram-Negative Bacteria

**DOI:** 10.3390/antibiotics11081115

**Published:** 2022-08-17

**Authors:** Catarina Bourgard, Diego Rodríguez-Hernández, Anastasia Rudenko, Carolin Rutgersson, Martin Palm, D. G. Joakim Larsson, Anne Farewell, Morten Grøtli, Per Sunnerhagen

**Affiliations:** 1Department of Chemistry and Molecular Biology, University of Gothenburg, S-405 30 Gothenburg, Sweden; 2Centre for Antibiotic Resistance Research (CARe), University of Gothenburg, S-405 30 Gothenburg, Sweden; 3Institute of Biomedicine, Department of Infectious Diseases, University of Gothenburg, S-413 46 Gothenburg, Sweden

**Keywords:** antibiotic resistance, antimicrobial activity, *Escherichia coli*, ESKAPE bacteria, dicationic compounds, sensitivity profiling

## Abstract

Antibiotic resistance among bacteria is a growing global challenge. A major reason for this is the limited progress in developing new classes of antibiotics active against Gram-negative bacteria. Here, we investigate the antibacterial activity of a dicationic bisguanidine-arylfuran, originally developed as an antitrypanosomal agent, and new derivatives thereof. The compounds showed good activity (EC_50_ 2–20 µM) against antibiotic-resistant isolates of the Gram-negative members of the ESKAPE group (*Klebsiella pneumoniae*, *Acinetobacter baumannii*, *Pseudomonas aeruginosa*, *Enterobacter* spp.) and *Escherichia coli* with different antibiotic susceptibility patterns, including ESBL isolates. Cytotoxicity was moderate, and several of the new derivatives were less cytotoxic than the lead molecule, offering better selectivity indices (40–80 for several ESKAPE isolates). The molecular mechanism for the antibacterial activity of these molecules is unknown, but sensitivity profiling against human ESKAPE isolates and *E. coli* collections with known susceptibility patterns against established antibiotics indicates that it is distinct from lactam and quinolone antibiotics.

## 1. Introduction

Bacterial resistance to antibiotics currently represents one of the biggest threats to global public health with more than 1 million people worldwide dying each year because of drug-resistant infections [1]. For this reason, the design and development of new classes of antibiotics are essential. Gram-negative bacteria have the highest resistance indices of all pathogenic bacteria, and development of new antibiotics to tackle them is urgently needed [2]. However, a recent report from the World Health Organization (WHO) [2] reveals a weak pipeline for antibiotics. The 60 products currently in development (50 antibiotics and 10 biologics) provide little advantage over existing treatments, and very few target Gram-negative bacteria.

The guanidine group, an organic base which is hydrophilic in nature, is commonly found in biologically active compounds, including antibiotics [3]. At physiological pH, the guanidine moiety is positively charged. The presence of this charge may lead to an electrostatic interaction between positively charged guanidine-containing compounds and, e.g., the negatively charged bacterial cell surface [3]. This immediate binding to the components of the cytoplasmic membrane or the cell wall causes the loss of biological functions of phospholipids, which can result in reduced membrane integrity. The resulting increase in membrane permeability leads to lysis and cell death [4]. Furthermore, the presence of a positive charge in guanidine derivatives may favor binding to intracellular targets, e.g., the minor groove of DNA [5].

The Pathogen Box (Medicines for Malaria Venture) is a further development of the Malaria Box collection [6]. This collection has been successfully used, e.g., to screen for small molecules active against pathogenic mycobacteria [7]. In the Pathogen Box, the dicationic 2,5-bis(2-chloro-4-guanidinophenyl)furan (**1**, MMV688179, Figure 1) caught our attention because it has two guanidinium groups in its structure [8]. The compound has an affinity for A/T-rich DNA [8]. Antifungal, antimycobacterial [8], and antiparasitic [9,10] activity has been reported for this compound, and it has been found to be active against the Gram-negative bacterium *Burkholderia pseudomallei* [11], the causative agent of the tropical disease melioidosis. However, its activity against *E. coli* and other pathogenic Gram-negative bacteria from the ESKAPE group, commonly causing serious infections around the world and exhibiting troublesome antibiotic resistance patterns, has not previously been examined. A variety of antimicrobial guanidine-containing compounds have been reported [3], targeting the bacterial envelope [12,13,14] or key bacterial proteins, such as DNA gyrase [15], lipid A and fatty acid biosynthesis enzymes [16], the bacterial cell division protein FtsZ4 [17], the NorA efflux pump [18], or targeting the ribosomal decoding rRNA site [19].

Because **1** has two guanidinium groups in its structure (Figure 1), we anticipate that bis-guanidine dicationic compounds bearing an arylfuran framework could be novel antibacterial agents. In this study, we describe the design, synthesis, and antibacterial evaluation of a series of dicationic derivatives with general structure **A** (Figure 1). The compounds were tested against a panel of five laboratory strains (one Gram-positive and four Gram-negative) and ten different Gram-negative bacteria isolates of human origin, representing *E. coli* and the ESKAPE group of species. In addition, we determined the cytotoxicity against MCF-7 and HepG2 human cell lines. To broadly characterize the mechanism of action of the new substances, we performed high-resolution microbial phenomics profiling of selected compounds and the known antibiotic cefotaxime (CTX) against two *E. coli* libraries (ECOR and ESBL) with characterized susceptibility patterns against established antibiotics.

## 2. Results

### 2.1. Chemistry

A set of compounds was prepared to address the key structure–activity relationships of the bis-arylfuran scaffold. Different substituents of the phenyl ring were investigated, as well as an isostere of the guanidine group. Asymmetric compounds were also explored (Figure 2). The first of these was a series of 2,5-bis(4-guanidino-aryl)furan derivatives which were synthesized (Appendix A) using the corresponding di-amino compounds **1b** and **3b**–**9b** as common precursors.

Initially, two different methodologies were tested to obtain 2,5-bis(4-nitroaryl)furans (**1a**, **3a**–**9a**): (1) a Suzuki cross-coupling reaction using the corresponding furan-2,5-diboronic acid pinacol ester with a substituted aryl bromide, and (2) a direct palladium-catalyzed arylation using furan with a substituted aryl bromide. In both cases, monoaryl furan was the major product obtained. This led us to use the synthetic approach previously reported by Stephens et al. [8,20] with some modifications. The synthesis of the amino compounds was achieved in two steps (Appendix A). Firstly, a Stille coupling using 2,5-bis(tri-*n*-butylstannyl)furan and a substituted 4-bromonitroarene was performed to form the corresponding 2,5-bis(4-nitrophenyl) furans (**1a**, **3a**–**9a**) in good to excellent yields (45–80%). From these intermediates (**1a, 3a**–**9a**), the nitro compounds were then reduced using iron powder with ammonium chloride to obtain the desired diamino compounds (**1b**, **3b**–**9b**) in excellent overall yields. The final transformation to obtain the target was carried out in two steps (Appendix A). The diamines were first reacted with Boc-protected *S*-methylthiourea in the presence of mercuric chloride (**1c**, **3c**–**9c**), followed by Boc-deprotection of the guanidine derivatives using 4 M HCl in dioxane. Ultimately, a good overall yield of the target compounds (**1**, **3**–**9**) was obtained. 

To investigate whether the activity is modulated by the cationic moiety when it is not directly attached to an aromatic system, we modified **3** by adding an additional carbon atom to extend the space between the aryl group and the guanidine moiety. This required the synthesis of the diamine **10b** (Appendix A), which was prepared in a two-step process involving palladium-catalyzed direct arylation using furan with 4-bromobenzonitrile, followed by reduction of the cyano group by treatment with lithium aluminum hydride. The bis-guanidine compound **10** was prepared (Appendix A) using the same synthetic strategy used for **1**.

Next, we exchanged the guanidinium residue of the bis-arylfuran scaffold for a bio-isostere. Here we chose to use the squaryldiamide moiety, as this group has been identified as a new potential bioisostere for unsubstituted guanidine functionality in peptidomimetics [21]. The 1,2-diaminocyclobutene-3,4-dione (squaryldiamide) derivatives were prepared using the diamine compounds through two synthetic steps (Appendix A). The diethyl squarate was first treated with the corresponding diamine compounds (**1b**, **3b**–**5b**) to displace one ethoxy residue and produce the corresponding intermediates **11a**–**14a**. These intermediates were then treated with ammonia to displace the second ethoxy group and produce the squaryldiamide derivatives **11**–**14** in good overall yields.

To modulate the activity of the bis-arylfuran scaffold, we synthesized asymmetric furan derivatives, replacing one of the aromatic rings with a residue containing an amide attached to an aromatic heterocycle. A series of 5-arylfuran-2-yl-indoline guanidine derivatives were synthesized. Appendix A summarizes the method used for the preparation of asymmetric furan guanidine compounds **16**–**20**. Like the guanidine compounds, the asymmetric derivatives required the corresponding amino or diamine compounds **16b**–**20b** as a common precursor. The preparation of these intermediary amines was performed in three steps starting with an amide reaction between 2-furoyl chloride and 5-nitroindoline or indoline to form the corresponding amides **2** and **15** in moderate yields. Then, an arylation reaction [21] was performed to add a substituted aryl bromide group to the 5-furanoyl amide derivatives. The bis and mono nitro compounds **16a**–**20a** were obtained in reasonable yield through this coupling reaction. Finally, the nitro compounds were reduced with iron to provide the desired amine compounds **16b**–**20b**. The asymmetric guanidine compounds **16**–**20** were prepared from the amine and di-amine (Appendix A) via the same general synthetic route used for **1**. Finally, the NMR analyses together with the HRMS studies confirm that all the synthesized compounds have a high degree of purity.

### 2.2. Evaluation of Cytotoxicity and Antimicrobial Activity against Gram-negative and Gram-positive Non-pathogenic Bacterial Strains 

The cytotoxicity of all compounds was evaluated using the human MCF-7 and HepG2 cell lines (Table 1; dose–response curves are shown in Appendix A). In general, all derivatives were moderately cytotoxic with an effective concentration of 50% (EC_50_) greater than 25 µM in both cell lines. Most of the new derivatives (**3**, **4**, **6**, **7**, **8**, **9**, **10**, **11**, **12**, **13**, **14**, **16**, **17**, **18**) were less cytotoxic than the lead (**1**). 

Cationic compounds **1**, **3**–**14**, and **16**–**20** were initially tested on a panel of five non-pathogenic bacterial strains, including both Gram-negative (*Escherichia coli*, *Pseudomonas putida*, *Pectobacterium carotovorum*, *Paraburkholderia caledonica*) and Gram-positive (*Bacillus subtilis*) species (Appendix A; dose–response curves are showed in Appendix A and S3). Ampicillin EC_50_ and EC_90_ values were used as a reference for *B.*
*subtilis* and *E. coli* strains (Appendix A).

Lead compound **1** bearing an electron-withdrawing Cl group on the phenyl ring and its isosteres **5**, **7**, and **8** bearing a CF_3_, CN, and COOCH_3_ group on the phenyl ring, had low antibacterial activity against all tested strains (Appendix A). Compounds bearing an electron donating group on the phenyl ring such as CH_3_ (**4**) or OCH_3_ (**6**) had moderate antibacterial activity. Both compounds had an EC_50_ of around 28.5 µM against Gram-negative *P. carotovorum* and around 14.4 µM against *P. putida* (Appendix A). Additionally, **4** had an EC_50_ of 8.2 µM against Gram-negative *P. caledonica* and was the most potent compound against the Gram-positive strain *B. subtilis* with an EC_90_ of 8.5 µM (Appendix A), which is 18 times more potent than ampicillin (Appendix A). When the phenyl ring in **4** was replaced with a pyridine ring (e.g., in **9**) the activity almost disappeared (Appendix A). Interestingly, **3** does not have any substituent on the aromatic ring and had good activity against the Gram-negative strains *P. putida* and *P. caledonica* (Appendix A). Compound **10** is characterized by a methylene between the guanidino group and the aromatic system with respect to **3** (Appendix A) and had a good EC_90_ value of 5 µM against *P. caledonica*. The squaryldiamide-based compounds (**11**–**14**) were the least active against all bacteria tested, indicating that the guanidino group is essential for antibacterial activity. Compounds **17** and **19** were the most potent against the Gram-negative *E. coli* (EC_90_ values of 4.3 and 2.5 µM, respectively; Appendix A), being 20- and 35-fold more potent than ampicillin (Appendix A). Compound **18** was the most active of the asymmetric series of compounds against Gram-negative *P. putida* and *P. caledonica* (Appendix A) and Gram-positive *B. subtilis* (Appendix A). 

### 2.3. Antibacterial Activity against ESKAPE and E. coli Isolates 

To evaluate the antimicrobial activity of the dicationic derivatives against clinically relevant bacteria, compounds **1**, **3**, **4**, **6**, **8**, **10**, **16** and **17**, which showed good activity against the non-pathogenic bacteria, were selected to be tested against 10 different Gram-negative isolates from the ESKAPE group and *E. coli*. The results are summarized in Table 2 and Table 3 and Figure 3 (dose–response curves are shown in Appendix A). Cefotaxime (CTX) was used as positive control (Appendix A). All compounds, except **10**, had moderate-to-good antibacterial activity against these isolates. 

For each species, two isolates with different susceptibility profiles to commonly used antibiotics were chosen. For CTX, the difference in sensitivity in each pair is obvious, ranging from 18× for the *A. baumannii* isolates to 1600× for K. pneumoniae (Appendix A). Likewise, resistance to meropenem is drastically different in each pair of isolates; from 8× for A. *baumannii* to 256× for K. pneumoniae, except for *Enterobacter* where the two isolates are equally resistant (Appendix A). The same is seen for ciprofloxacin resistance for *E. coli* (8000×), *K. pneumoniae* (1000×), and P. *aeruginosa* (32×) (Table S3). By contrast, for sensitivity to all the bisguanidine-arylfuran compounds, the difference in each pair was less than two-fold for all species, except for *Enterobacter* (7× difference). This clearly demonstrates that the antibacterial efficacy of the new diaryl compounds is not diminished by the mechanisms that cause resistance against established antibiotics.

Compounds **1** and **8**, which carry electron-withdrawing groups, had better activity than compounds with electron-donating groups (e.g., **4** and **6**) in all the strains evaluated. Compound **1** had EC_90_ values below 10 µM against all the ESKAPE isolates evaluated, regardless of their resistance profile to established antibiotics (Table 2 and Appendix A). Thus, in the case of the tested *P. aeruginosa* isolate CCUG #59347, the EC_90_ of 7.4 µM was 228 times lower than that of CTX (EC_90_ = 1684 µM), but for isolate CCUG #17619 the EC_90_ of 6.7 µM was only 2.7 times lower than that of CTX (EC_90_ = 18.4 µM) (Table 2). Compound **1** had an EC_90_ of 3.8 µM against *E. coli* CCUG #67180, 48 times more potent than CTX (EC_90_ = 183.1 µM), but against *E. coli* CCUG #17620 **1** (EC_90_ = 6.3 µM) was 12 times less potent than CTX (EC_90_ = 0.3 µM). Similarly, **1** was approximately 60 times more potent than CTX against the antibiotic-tolerant isolate *K. pneumoniae* CCUG #58547 but 11 times less potent against CCUG #225T; 108 times more potent against *A. baumannii* CCUG #57035 but only 7 times more effective against CCUG #57250 (Table 2). Compound **8** was the most potent against the more antibiotic-tolerant *Enterobacter* isolate (CCUG #58962) with an EC_90_ of 8.7 µM, 18 times more potent than CTX (EC_90_ = 158.3 µM). 

Compound **3**, without substituents in the aromatic system, had the best potency in several isolates, and was more potent than CTX in most cases. For instance, against the comparatively CTX-tolerant *A. baumannii* CCUG #57035 and #57250 (Appendix A), the EC_90_ of 2.4 µM of **3** was 283 times and 20 times lower, respectively, than those for CTX (EC_90_ = 680.6 and 48 µM). This results in a selectivity index of approximately 70 for both isolates (Table 3). In line with the results for **1**, **3** was 208, 57 and 22 times more potent than CTX against the antibiotic-tolerant isolates *P. aeruginosa* CCUG #59347 (EC_90_ = 8.1 µM), *K. pneumoniae* CCUG #58547 (EC_90_ = 3.1 µM), and *E. coli* CCUG #67180, (EC_90_ = 2.2 µM), respectively (Table 2). For the corresponding more antibiotic-susceptible isolates, *P. aeruginosa* CCUG #17619, *K. pneumoniae* CCUG #225T, and *E. coli* CCUG #17620, the analogous ratios for **3** were 1.6, 0.1 and 0.15 relative to CTX.

When comparing the activity of **3** with **10**, the addition of a carbon between the aromatic system and the guanidino moiety did not have an impact on the activity against the ESKAPE isolates. Finally, the asymmetric **16** and **17** had moderate activity against all the strains evaluated. Compound **3** displayed the highest selectivity index, between 13 and 83 across all ESKAPE isolates (Table 3). This represents an improvement over **1** by a factor between 1.7 and 9, achieved in all cases through weaker cytotoxicity. Therefore, it is important to analyze the differences in cytotoxicity when evaluating the practical antibiotic potential of the new compounds. 

### 2.4. Sensitivity Profiling against Two Collections of E. coli Strains with Defined Antibiotic Susceptibility Patterns

As an approach to finding the mechanism of action of the new substances, we performed high-resolution microbial phenomics profiling of selected compounds **1**, **4**, **6**, **10**, and **16** and the known antibiotic CTX against two *E. coli* libraries with known patterns of antibiotic susceptibility. The *E. coli* reference collection (ECOR) contains 72 strains isolated from a wide variety of environments and geographical locations [22] including representatives of the seven *E. coli* phylogroups [23]. Eighteen of the strains are resistant to one known antibiotic (12 antibiotics tested) and 14 are resistant to two antibiotics, with resistance to sulfisoxazole, tetracycline and streptomycin being the most common [24]. A set of 96 extended spectrum beta-lactamase (ESBL) strains were isolated at Sahlgrenska University Hospital in Gothenburg, Sweden between 2011 and 2012, some of which have been previously published [25,26]. These are likely mostly closely related since they were isolated from a small patient population and are in large part uncharacterized except for their identification as ESBL *E. coli* strains. The results of these experiments are summarized in Figure 4 and Appendix A. In Figure 4, we see the growth yield of the strains in these collections upon exposure to diaryl compounds relative to our reference strain (*E. coli* ATCC #25922) and normalized for growth without any added compound. Primarily, there was no widespread resistance to the diaryl compounds in either strain set. As expected, most ESBL strains are highly resistant to the lactam CTX, but are no more resistant to any diaryl compound than the control strain (Figure 4A). Importantly, the profiles across the strains of CTX resistance do not covary with the resistance profiles for any of the diaryls (Figure 4A,B). Strains strongly resistant to CTX showed normal sensitivity to the diaryl compounds (e.g., GU1114, GU1078), whereas strains that were particularly sensitive to several of the diaryls display normal CTX sensitivity (GU1117, GU1068, GU2320). 

## 3. Discussion

### 3.1. Efficiency of Compounds, Relationship between Structure and Function

For decades, no new classes of antibiotics effective against Gram-negative bacteria have reached the market. The current rise in antibiotic resistance among such bacteria therefore threatens to deplete the remaining clinical treatment options. Especially troubling is the fact that acquired co-resistance to several antibiotics in one bacterial population (multidrug resistance) is common [28]. Infections by some Gram-negative species are particularly problematic to treat with antibiotics. It is therefore promising that the human isolates of *E. coli* and ESKAPE species tested are sensitive to the new molecules presented here. The sensitivity patterns we observe across bacterial strains, which do not correlate with their sensitivity to established antibiotics, also indicate that the mechanism of action of diaryl compounds may be different from those. 

It is noteworthy that the new compounds show good activity against *A. baumannii*. This opportunistic pathogen is notoriously difficult to treat with antibiotics, as it displays high level intrinsic resistance to many antibiotics [29], which has been attributed at least in part to abundant membrane-bound export pumps [30]. It will be important to identify which properties allow these molecules to escape such barriers to cellular uptake. Future work could examine, for instance, what distinguishes **1** and **3**, which both show a particularly high efficiency against *A. baumannii* (Table 2), from the other related compounds in the design series. 

*K. pneumoniae* is strongly implicated in nosocomial infections and accumulates multiple plasmid-borne antibiotic-resistance genes [31], making this bacterial species a major medical problem. It is therefore encouraging that several of the compounds presented here (e.g., **1**, **3**, **4**, **6**, **8**, **16**) are effective against multidrug resistant *K. pneumoniae* (Table 2 and S3), with selectivity indices between 11 and 24 (**3**, **6**, **8**, **16**; Table 3). *P. aeruginosa*, in addition to widespread antibiotic resistance, has a high propensity to form biofilms, adding further difficulty to the clinical treatment of infections [32]. Here, we have only examined planktonic *P. aeruginosa*, however, we see that **1** and **3** were more effective than CTX against both isolates of *P. aeruginosa* (Table 2).

Compound **3** was more effective than the lead compound against some of the bacterial isolates (Appendix A) and is less cytotoxic than **1** (Table 1). Even in cases where **3** did not show more antibacterial activity than **1** (*K. pneumoniae*, *P. aeruginosa*, *Enterobacter*), this combination results in better selectivity indices. Therefore, **3**, which lacks substituents in the aromatic systems, represents a favorable compromise in situations where the chlorine substituents in **1** (Figure 2), which may confer toxicity, have been eliminated (Table 3). Compound **8** also has better selectivity indices than **1**, against *K. pneumoniae*, *A. baumannii*, and *Enterobacter* (Table 3). This molecule has methoxycarbonyl groups replacing the chlorine substituents in the lead compound (Figure 2). The improved selectivity indices of **8** over **1** is mainly due to reduced cytotoxicity (Table 1, Table 2 and Table 3). By contrast, **4** and **6**, which carry electron-donating methyl or methoxy groups on the phenyl ring (Figure 2), displayed greater activity than **1** against several laboratory strains (Appendix A). However, their selectivity indices were less favorable because of higher cytotoxicity (Table 1 and Table 3). 

Interestingly, antibacterial compounds carrying an aminoguanidine group have been reported to potentiate norfloxacin in *Staphylococcus aureus*, suggesting a possibility to develop the guanidine-containing molecules reported here as co-drugs [18].

### 3.2. Mechanism of Action

The mechanism for the antibacterial effect of this group of compounds is not clear. The lead compound and derivatives were originally designed to target A/T-rich DNA sequences [8]. However, **1** is comparably effective against the eukaryotic parasites *Plasmodium falciparum* (EC_50_ for erythrocyte stage = 590 nM) [10], which has a genome with an exceptionally low G/C content of 19.8% [33], and against *Trypanosoma cruzi* (290 nM) [34], which has a G/C content of 51.0% [35]. Additionally, there is no obvious correlation between the genomic G/C content of the bacterial species examined here and their sensitivity to the compounds used in this work (Appendix A). And while these compounds were thought to selectively bind in the DNA minor groove [8] there is no structural difference between prokaryotic and eukaryotic DNA that could account for an antibacterial effect. A DNA-binding compound would also be suspected to be genotoxic. However, we are not aware of any reports on mutagenic or carcinogenic activity of dicationic bisguanidine-arylfurans. Together, this argues against DNA-binding as the main mechanism for the observed antibacterial activity.

The mode of action of a compound can be probed indirectly by comparing with the sensitivity of bacteria to established antibiotics with known action mechanisms. It is evident from analyzing the *E. coli* strains (Figure 4) that the resistance profiles for the diaryl compounds across the strain collections are distinct from that of CTX, the lactam compound used here as a reference. This conclusion is also supported by the resistance pattern of the ESKAPE isolates (Table 2 and Appendix A), where the aryl compounds were equally effective against isolates that are resistant or sensitive to established antibiotics. CTX is a lactam in the cephalosporin subgroup. Meropenem is also a lactam compound, though belonging to the carbapenem subgroup, while ciprofloxacin is a quinolone and acts by interfering with bacterial DNA replication. This argues that whatever mechanisms underlie the resistance of these isolates of ESKAPE species and *E. coli* to established antibiotics, they are not effective against the diaryl compounds in this study. It is therefore plausible that the mode of action of the diaryl compounds is distinct from commonly used antibiotics, at least from lactam compounds and quinolones. 

Given all the above, it also has to be considered that a small molecule may interact with multiple cellular targets at any given time. Which of these will result in the biologically relevant effect may in addition vary with extrinsic and intrinsic factors acting on the cell, and on uptake and intracellular localization.

## 4. Materials and Methods

### 4.1. General Experimental Information for Synthesis and Compound Characterization

General reagent and solvents for the synthesis of compounds were purchased from commercial sources and used as supplied, unless otherwise stated. 

Purification by flash column chromatography was performed on a Selekt (Biotage, Uppsala, Sweden) automated instrument with Sfär KP-amino D or Sfär silica D cartridges (Biotage, Uppsala, Sweden), mobile phase consist of pentane (solvent A) and ethyl acetate (solvent B). The final compounds were purified by reverse phase flash column chromatography performed on an Isolera (Biotage, Uppsala, Sweden) automated instrument with Sfär C18 D cartridges (Biotage, Uppsala, Sweden), mobile phase consist of water (solvent A) and acetonitrile (solvent B). The standard gradient consisted of *x%* solvent B for 1 columns volume, *x%* to *y%* B for 10 column volumes, and then *y%* B for 2 column volumes. *x* and *y* are defined in the characterization section of the compound the interest. 

All NMR spectra (^1^H and ^13^C) were recorded on a Varian 400 MHz spectrometer (Varian, Palo Alto, CA, USA) at 25 °C. Samples were dissolved (0.5 mL) in deuterated chloroform, methanol or dimethylsulfoxide (CDCl_3_, CD_3_OD, DMSO-*d*_6_). The residual solvent peaks specific to that to the deuterated solvent was used as an internal reference; CDCl_3_: 7.26 ppm (^1^H NMR) and 77.20 ppm (^13^C NMR); CD_3_OD: 3.31 ppm (^1^H NMR) and 49.00 ppm (^13^C NMR); DMSO-*d*_6_: 2.50 ppm (^1^H NMR) and 39.52 ppm (^13^C NMR). Data are presented as follows: chemical shift in ppm, multiplicity (br = broad, s = singlet, d = doublet, dd = doublet of doublet, t = triplet, q = quartet, m = multiplet), coupling constants in Hz and integration. High resolution mass spectra (HRMS) were recorded on an Agilent 1290 infinity LC system tandem to an Agilent 6520 Accurate Mass Q-TOF spectrometer (Agilent, Santa Clara, CA, USA).

### 4.2. General Procedure A 

In a sealed 20 mL microwave vial, 2,5-Bis-(trimethylstannyl)furan (0.5 mmol), aryl bromide (1.0 mmol) and tetrakis(triphenyphosphine)-palladium(0) (0.025 mmol) in anhydrous dimethylformamide (10 mL) was evacuated and backfilled with N_2_ (×3) and heated for 14 h at 100 °C. Upon cooling, the mixture was filtered through Celite, the Celite rinsed with chloroform, and the residue was reduced under vacuum. Then 100 mL of chloroform, and 50 mL of 10% aqueous potassium fluoride was added and the mixture was stirred at room temperature for 0.5 h. The organic layer was separated and dried over sodium sulfate, filtered, and concentrated under reduced pressure. The crude product was purified by flash chromatography (25 g Sfär silica D cartridge, 15–75% B, R*_f_* = 12 column volumes) to give the desired compounds.

*2,5-Bis(4-nitro-2-chlorophenyl)furan (**1a**).* Following general procedure A, 1-bromo-2-chloro-4-nitrobenzene was used as aryl bromide. Afforded the title compound as an orange solid, yield 45%. ^1^H NMR (400 MHz, DMSO-*d*_6_, *δ*, ppm): 8.45 (d, *J* = 2.2 Hz, 2H), 8.38 (d, *J* = 8.8 Hz, 2H), 8.32 (dd, *J* = 8.8; 2.2 Hz, 2H), 7.72 (s, 2H). Data is consistent with that previously reported [8].

*2,5-Bis(4-nitrophenyl)furan (**3a**).* Following general procedure A, 1-bromo-4-nitrobenzene was used as aryl bromide. Afforded the title compound as orange solid, yield 33%. ^1^H NMR (400 MHz, DMSO-*d*_6_, *δ*, ppm): 8.35 (d, *J* = 9.0 Hz, 4H), 8.18 (d, *J* = 9.0 Hz, 4H), 7.44 (s, 2H). Data is consistent with that previously reported [8].

*2,5-Bis(4-nitro-2-methylphenyl)furan (**4a**).* Following general procedure A, 1-bromo-2-methyl-4-nitrobenzene was used as aryl bromide. Afforded the title compound as tan solid, yield 62.2%. ^1^H NMR (400 MHz, DMSO-*d*_6_, *δ*, ppm): 8.27 (d, *J* = 2.0 Hz, 2H), 8.19 (dd, *J* = 8.9; 2.0 Hz, 2H), 8.15 (d, *J* = 8.9 Hz, 2H), 7.36 (s, 2H), 2.71 (s, 6H). Data is consistent with that previously reported [8].

*2,5-Bis(4-nitro-2-trifluoromethylphenyl)furan (**5a**).* Following general procedure A, 2-bromo-5-nitrobenzotrifluoride was used as aryl bromide. Afforded the title compound as golden solid, yield 15%. ^1^H NMR (400 MHz, DMSO-*d*_6_, *δ*, ppm): 8.63 (d, *J* = 8.7 Hz, 2H), 8.59 (s, 2H), 8.25 (d, *J* = 8.7 Hz, 2H), 7.40 (s, 2H). Data is consistent with that previously reported [8].

*2,5-Bis(2-methoxy-4-nitro-phenyl)furan (**6a**).* Following general procedure A, 1-bromo-2-methoxy-4-nitrobenzene was used as aryl bromide. Afforded the title compound as orange solid, yield 80%. ^1^H NMR (400 MHz, DMSO-*d*_6_, *δ*, ppm): 8.27 (d, *J* = 8.7 Hz, 2H), 7.97 (dd, *J* = 8.5; 2.0 Hz, 2H), 7.22 (d, *J* = 2.0 Hz, 2H), 7.41 (s, 2H), 4.11 (s, 6H). Data is consistent with that previously reported [8].

*2,5-Bis(2-cyano-4-nitro-phenyl)furan (**7a**).* Following general procedure A, 2-bromo-4-nitrobenzonitrile was used as aryl bromide. Afforded the title compound as pale yellow solid, yield 46%. ^1^H NMR (400 MHz, DMSO-*d*_6_, *δ*, ppm): 8.89 (d, *J* = 2.1 Hz, 2H), 8.63 (dd, *J* = 8.6; 2.1 Hz, 2H), 8.43 (d, *J* = 8.6 Hz, 2H), 7.86 (s, 2H). 

*2,5-Bis(4-nitro-2-(methyl carboxy)-phenyl)furan (**8a**).* Following general procedure A, methyl 2-bromo-5-nitrobenzoate was used as aryl bromide. Afforded the title compound as orange solid, yield 68%. ^1^H NMR (400 MHz, DMSO-*d*_6_, *δ*, ppm): 8.48–8.45 (m, 4H), 8.06 (m, 2H), 7.31 (s, 2H), 3.83 (s, 6H). 

*2,5-Bis(2-methyl-6-nitropyridin-3-yl)furan (**9a**).* Following general procedure A, 3-bromo-2-methyl-6-nitropyridine was used as aryl bromide. Afforded the title compound as tan solid, yield 63%. ^1^H NMR (400 MHz, DMSO-*d*_6_, *δ*, ppm): 8.36 (d, *J* = 6.6 Hz, 2H), 8.33 (d, *J* = 6.6 Hz, 2H), 6.98 (s, 2H), 2.95 (s, 6H).

### 4.3. General Procedure B 

To a solution of 2,5-Bis(4-nitrophenyl)furan derivatives (**1a**, **3a**–**9a**) (0.3 mmol) in THF (3 mL) and EtOH (3 mL), ammonium chloride (3 mL, 0.3 M) and iron (1.75 mmol) were added After stirring at 60 °C for 4 h, the reaction was allowed to cool to room temperature and the heterogeneous mixture filtered through Celite and the Celite was rinsed with ethyl acetate. The solution was concentrated to half-volume, then diluted with ethyl acetate (20 mL) and washed with sodium hydroxide solution (1 M, 20 mL). The organic layer was separated, the aqueous phase was extracted with ethyl acetate (2×), the combined organic phases dried over sodium sulfate, filtered and the solvent evaporated. The crude product was purified by flash chromatography (11 g Sfär KP-amino D cartridge, 15–90% B, R*_f_* = 10 column volumes) to give the desired compounds.

*2,5-Bis(4-amino-2-chlorophenyl)furan (**1b**).* Compound **1a** was reacted according to General Procedure B, yield 82%, red solid. ^1^H NMR (400 MHz, CDCl_3_, *δ*, ppm): 7.58 (d, *J* = 8.5 Hz, 2H), 6.83 (s, 2H), 6.69 (d, *J* = 2.2 Hz, 2H), 6.62 (dd, *J* = 8.5; 2.2 Hz, 2H), 5.64 (brs, 4H). Data is consistent with that previously reported [8].

*2,5-Bis(4-amino-phenyl)furan (**3b**).* Compound **3a** was reacted according to General Procedure B, yield 96%, red solid. ^1^H NMR (400 MHz, CDCl_3_, *δ*, ppm): 7.53 (d, *J* = 8.7 Hz, 4H), 6.70 (d, *J* = 8.7 Hz, 4H), 6.49 (s, 2H), 3.72 (br s, 4H). Data is consistent with that previously reported [8].

*2,5-Bis(4-amino-2-methylphenyl)furan (**4b**).* Compound **4a** was reacted according to General Procedure B, yield 76%, red solid. ^1^H NMR (400 MHz, DMSO, *δ*, ppm): 7.41 (d, *J* = 8.0 Hz, 2H), 6.51 (m, 4H), 6.47 (s, 2H), 5.20 (s, 4H), 2.38 (s, 6H). Data is consistent with that previously reported [8].

*2,5-Bis(4-amino-2-trifluoromethylphenyl)furan (**5b**).* Compound **5a** was reacted according to General Procedure B, yield 96%, red solid. ^1^H NMR (400 MHz, CDCl_3_, *δ*, ppm): 7.59 (d, *J* = 8.4 Hz, 2H), 7.00 (d, *J* = 2.4 Hz, 2H), 6.81 (dd, *J* = 8.4; 2.4 Hz, 2H), 6.62 (s, 2H), 3.90 (br s, 4H). Data is consistent with that previously reported [8].

*2,5-Bis(4-amino-2-methoxyphenyl)furan (**6b**).* Compound **6a** was reacted according to General Procedure B, yield 90%, dark red solid. ^1^H NMR (400 MHz, CDCl_3_, *δ*, ppm): 7.73 (d, *J* = 8.3, Hz, 2H), 6.77 (s, 2H), 6.36 (dd, *J* = 8.3; 1.9 Hz, 2H), 6.28 (d, *J* = 1.9 Hz, 2H), 3.88 (s, 6H), 3.73 (br s, 4H). Data is consistent with that previously reported [8].

*2,5-Bis(4-amino-2-cyanophenyl)furan (**7b**).* Compound **7a** was reacted according to General Procedure B, yield 75%, yellow solid. ^1^H NMR (400 MHz, DMSO-*d*_6_, *δ*, ppm): 7.82 (d, *J* = 9.2 Hz, 2H), 7.07 (s, 2H), 7.05–7.02 (m, 4H), 5.39 (br s, 4H). 

*2,5-Bis(4-amino-2-(methyl carboxy)-phenyl)furan (**8b**).* Compound **8a** was reacted according to General Procedure B, yield 95%, orange solid. ^1^H NMR (400 MHz, CDCl_3_, *δ*, ppm): 7.32 (d, *J* = 9.0 Hz, 2 H), 6.73–6.70 (m, 4H), 6.39 (s, 2H), 5.58 (br s, 4H), 3.69 (s, 6H). 

*2,5-Bis(6-amino-2-methylpyridin-3-yl)furan (**9b**).* Compound **9a** was reacted according to General Procedure B, yield 88%, orange solid. ^1^H NMR (400 MHz, CDCl_3_, *δ*, ppm): 7.68 (d, *J* = 8.5 Hz, 2 H), 6.54 (s, 2H), 6.37 (d, *J* = 8.5 Hz, 2 H), 5.07 (br s, 4H), 2.46 (s, 6H).

*2,5-Bis-(4-cyanophenyl)**furan (**10a**)* A sealed 20 mL microwave vial containing 4-bromobenzonitrile (1.5 mmol), furan (4.5 mmol), potassium acetate (3.0 mmol) and palladium(II) acetate (0.015 mmol) in dimethylacetamide (5 mL) was evacuated and backfilled with N_2_ (×3), and heated for 20 h at 150 °C. Upon cooling to room temperature, the residue filtered through Celite and the Celite rinsed with ethyl acetate. The solution was washed with water and brine, dried over sodium sulfate, filtered, and concentrated under reduced pressure. The crude product was purified by flash chromatography (25 g Sfär silica D cartridge, 15–75% B, R*_f_* = 12 column volumes) to give the compound as a dark yellow solid, yield 29%. ^1^H NMR (400 MHz, DMSO-*d*_6_, *δ*, ppm): 8.03 (d, *J* = 8.5 Hz, 4H), 7.90 (d, *J* = 8.5 Hz, 4H), 7.42 (s, 2H). Data is consistent with that previously reported [36]. 

*2,5-Bis(4-aminomethylphenyl)furan (**10b**)* To a solution of 2,5-Bis(4-cyanophenyl)furan (**10a**) (0.5 mmol) in THF (10 mL) was added to Lithium aluminum hydride (4.7 mmol) in tetrahydrofuran (15 mL) at 0 °C, and stirred at room temperature overnight. Sodium hydroxide (5 mL, 10% solution) was added and stirred at 0 °C after 30 min, water (10 mL) was added to give a granular precipitate. The mixture was filtered, and the precipitate washed copiously with ether and dried in vacuo to give the product as a pale-yellow solid, yield 60%, that were used without further purification. ^1^H NMR (400 MHz, DMSO-*d*_6_, *δ*, ppm): 7.74 (d, *J* = 8.1 Hz, 4H), 7.40 (d, *J* = 8.1 Hz, 4H), 7.00 (s, 2H), 3.73 (s, 4H). Data is consistent with that previously reported [37].

### 4.4. General Procedure C 

A solution of 2,5-Bis(4-aminoaryl)furan derivatives (**1b**, **3b**–**9b**) or 2,5-Bis(4-aminomethylphenyl)furan (**10b**) (0.10 mmol) mercury (II) chloride (0.21 mmol), 1,3-bis(*tert*-butoxycarbonyl)-2-methyl-2-thiopseudourea (0.19 mmol) and triethylamine (0.48 mmol).in either dichloromethane or dimethylformamide (5 mL) was stirred at 0 °C for 1 h and then at room temperature for 18 h. Then, the reaction mixture was diluted with ethyl acetate filtered through Celite and the Celite rinsed with ethyl acetate. The organic phase was washed with water (2×), and brine (2×), dried over sodium sulfate, filtered, and concentrated under reduced pressure. The crude product was purified by flash chromatography (10 g Sfär D cartridge, 15–80% B, R*_f_* = 10 column volumes) to give the desired compounds.

*2,5-Bis(2-chloro-4-N,N′-di-(tert-butoxycarbonyl)guanidinophenyl)furan (**1c**).* Compound **1b** was reacted according to General Procedure C**.** Afforded the title compound as pale-yellow solid, yield 73%. ^1^H NMR (400 MHz, CDCl_3_, *δ*, ppm): 11.61 (brs, 2H), 10.45 (brs, 2H), 7.90 (d, *J* = 8.7, Hz, 2H), 7.81 (d, *J* = 2.1 Hz, 2H), 7.65 (dd, *J* = 8.7; 2.1, 2H), 7.19 (s, 2H), 1.54 (s, 18H), 1.53 (s, 18H). Data is consistent with that previously reported [8].

*2,5-Bis(4-N,N′-di**-*(*tert-butoxycarbonyl*)*guanidinophenyl)furan (**3c**).* Compound **3b** was reacted according to General Procedure C**.** Afforded the title compound as pale yellow solid yield 63%. ^1^H NMR (400 MHz, CDCl_3_, *δ*, ppm): 11.65 (br s, 2H), 10.32 (br s, 2H), 7.60 (br d, *J* = 8.6 Hz, 4H), 7.33–7.29 (m, 4H), 7.10 (m, 2H), 1.53 (s, 18H), 1.50 (s, 18H). Data is consistent with that previously reported [8].

*2,5-Bis(2-methyl-4-N,N′-di-**(**tert-butoxycarbonyl*)*guanidinophenyl)furan (**4c**).* Compound **4b** was reacted according to General Procedure C**.** Afforded the title compound as pale-yellow solid, yield 56%. ^1^H NMR (400 MHz, CDCl_3_, *δ*, ppm): 11.63 (br s, 2H), 10.35 (br s, 2H), 7.74 (d, *J* = 8.6 Hz, 2H), 7.64 (m, 2H), 7.42 (br s, 2H), 6.61 (br s, 2H), 2.53 (s, 6H), 1.54 (s, 18H), 1.52 (s, 18H). Data is consistent with that previously reported [8].

*2,5-Bis(2-trifluoromethyl-4-N,N′-di**-(**tert-butoxycarbonyl**)guanidinophenyl)furan (**5c**).* Compound **5b** was reacted according to General Procedure C. Afforded the title compound as yellow/orange solid, yield 81%. ^1^H NMR (400 MHz, CDCl_3_, *δ*, ppm): 11.61 (br s, 2H), 10.55 (br s, 2H), 8.03 (dd, *J* = 8.6; 2.2, Hz, 2H), 7.95 (d, *J* = 2.2 Hz, 2H), 7.84 (d, *J* = 8.5, 2H), 6.79 (s, 2H), 1.55 (s, 18H), 1.52 (s, 18H). Data is consistent with that previously reported [8].

*2,5-Bis(2-methoxy-4-N,N′-di**-(**tert-butoxycarbonyl**)guanidinophenyl)furan (**6c**).* Compound **6b** was reacted according to General Procedure C Afforded the title compound as red solid, yield 93%. ^1^H NMR (400 MHz, CDCl_3_, *δ*, ppm): 11.58 (br s, 2H), 10.38 (br s, 2H), 7.88 (d, *J* = 8.5, Hz, 2H), 7.60 (br s, 2H), 7.16 (br d, *J* = 8.5, 2H), 6.97 (s, 2H), 3.96 (s, 6H), 1.54 (s, 18H), 1.51 (s, 18H). Data is consistent with that previously reported [8].

*2,5-Bis(2-cyano-4-N,N′-di**-(**tert-butoxycarbonyl**)guanidinophenyl)furan (**7c**).* Compound **7b** was reacted according to General Procedure C. Afforded the title compound as pale yellow solid, yield 94%. ^1^H NMR (400 MHz, CDCl_3_, *δ*, ppm): 11.61 (br s, 2H), 10.57 (br s, 2H), 8.14 (br s, 2H), 8.06 (d, *J* = 8.5, 2H), 6.68 (s, 2H), 2.70 (s, 6H), 1.55 (s, 18H), 1.54 (s, 18H). 

*2,5-Bis(2-(methylcarboxy)-4-N,N′-di-(tert-butoxycarbonyl)guanidinophenyl)furan (**8c**).* Compound **8b** was reacted according to General Procedure C. Afforded the title compound as orange solid, yield 95%. ^1^H NMR (400 MHz, CDCl_3_, *δ*, ppm): 11.60 (br s, 2H), 10.50 (br s, 2H), 8.00 (dd, *J* = 8.6; 2.3, 2H), 7.76 (d, *J* = 2.3, 2H), 7.64 (br d, *J* = 8.6, 2H), 3.78 (s, 2H), 6.63 (s, 2H), 1.53 (s, 18H), 1.51 (s, 18H). 

*2,5-Bis(2-(methyl)-6-N,N′-di-*(*tert*-*butoxycarbonyl)guanidinopiridin-3-yl)furan (**9c**).* Compound **9b** was reacted according to General Procedure C. Afforded the title compound as tan solid, yield 31%. ^1^H NMR (400 MHz, CDCl_3_, *δ*, ppm): 11.54 (br s, 2H), 10.83 (br s, 2H), 8.31 (br s, 2H), 7.87 (d, *J* = 8.8, 2H), 7.84 (br d, *J* = 8.8, 2H), 7.30 (s, 2H), 1.54 (s, 18H), 1.52 (s, 18H).

*2,5-Bis(4-N,N′-di-*(*tert*-*butoxycarbonyl)guanidinomethylphenyl)furan (**10c**).* Compound **10b** was reacted according to General Procedure C. Afforded the title compound as yellow oil, yield 76%. ^1^H NMR (400 MHz, CDCl_3_, *δ*, ppm): 11.55 (br s, 2H), 8.61 (t, *J* = 4.9 Hz, 2H), 7.70 (d, *J* = 8.1 Hz, 4H), 7.34 (d, *J* = 8.1 Hz, 4H), 6.71 (s, 2H), 4.64 (d, *J* = 4.9 Hz, 4H), 1.51 (s, 18H), 1.47 (s, 18H).

### 4.5. General Procedure D 

To a solution of 2,5-Bis(4-*N,N’*-di-(tert-butoxycarbonyl)guanidinophenyl)furan derivatives (**1c**, **3c**–**9c**) or 2,5-Bis(4-*N,N*′-di-(tert-butoxycarbonyl)guanidinomethylphenyl)furan (**10c**) (0.05 mmol) in dichloromethane (1 mL) was added HCl (4 M in dioxane, 2 mL). The solution was stirred at room temperature for 16 h. The mixture was concentrated under reduced pressure and the residue was purified by reverse phase flash chromatography (6 g Sfär C18 D cartridge, 0–45% B, R*_f_* = 10 column volumes) to give the desired compounds.

*2,5-Bis(2-chloro-4-guanidinophenyl)furan (**1**).* Compound **2c** was reacted according to General Procedure D, yield 89%, tan solid. ^1^H NMR (400 MHz, CD_3_OD, *δ*, ppm): 8.11 (d, *J* = 8.6 Hz, 6H), 7.51 (dt, *J* = 2.2 Hz, 2H), 7.38 (br dd, *J* = 8.6; 2.2 Hz, 2H), 7.35 (br s, 2H). ^13^C NMR (100 MHz, CD_3_OD, *δ*, ppm): 157.90, 150.71, 136.50, 132.09, 130.24, 128.61, 128.15, 124.84, 114.52. Data is consistent with that previously reported [8].

*2,5-Bis(guanidinophenyl)furan (**3**).* Compound **3c** was reacted according to General Procedure D, yield 85%, light brown solid. ^1^H NMR (400 MHz, CD_3_OD, *δ*, ppm): 7.96 (d, *J* = 8.8 Hz, 4H), 7.42 (d, *J* = 8.8 Hz, 4H), 7.03 (s, 2H). ^13^C NMR (100^1^H NMR (400 MHz, DMSO-*d*_6_, *δ*, ppm): 158.06, 15410.13, (s, 2H), 7.56 (brs, 6H), 7.44 (t, *J* = 7. Hz, 4H), 7.29 (t, *J* = 7.5 Hz, 2H), 7.23 (d, *J* = 7.6 Hz, 4H). ^13^C NMR (100 MHz, DMSO-*d*_6_, *δ*, ppm): 156.04, 135.22, 131.1726, 129.66, 126.86, 126.19, 109.53.32, 124.25. Data is consistent with that previously reported [8].

*2,5-Bis(4-guanidino-2-methylphenyl)furan (**4**).* Compound **4c** was reacted according to General Procedure D, yield 90.2%, light brown solid. ^1^H NMR (400 MHz, CD_3_OD, *δ*, ppm): 7.95 (d, *J* = 8.1 Hz, 2H), 7.29 (m, 4H), 6.93 (s, 2H), 2.67 (s, 6H). ^13^C NMR (100 MHz, CD_3_OD, *δ*, ppm): 158.01, 153.45, 137.74, 135.15, 130.41, 129.34, 128.89, 123.91, 112.50, 22.25. Data is consistent with that previously reported [8]. 

*2,5-Bis(4-guanidino-2-trifluoromethylphenyl)furan (**5**).* Compound **5c** was reacted according to General Procedure D, yield 60%, orange solid. ^1^H NMR (400 MHz, DMSO-*d*_6_, *δ*, ppm): 10.53 (s, 2H), 7.93 (d, *J* = 8.6 Hz, 2H), 7.87 (br s, 6H), 7.70 (br s, 2H), 7.65 (d, *J* = 8.6 Hz, 2H), 7.00 (s, 2H). ^13^C NMR (100 MHz, DMSO-*d*_6_, *δ*, ppm): 156.34, 150.63, 136.68, 131.84, 128.02, 126.51, 126.20, 125.70, 122.52, 113.03. Data is consistent with that previously reported [8].

*2,5-Bis(4-guanidino-2-methoxyphenyl)furan (**6**).* Compound **6c** was reacted according to General Procedure D, yield 90%, red solid. ^1^H NMR (400 MHz, CD_3_OD, *δ*, ppm): 8.02 (d, *J* = 7.8 Hz, 2H), 7.07 (s, 2H), 7.02 (d, *J* = 2.0 Hz, 2H), 6.99 (dd, *J* = 8.3; 2.0 Hz, 2H), 4.00 (s, 6H). ^13^C NMR (100 MHz, CD_3_OD, *δ*, ppm): 157.92, 157.80, 149.50, 135.71, 127.56, 120.06, 118.06, 114.01, 109.49, 56.24. Data is consistent with that previously reported [8]. 

*2,5-Bis(2-cyano-4-guanidinophenyl)furan (**7**).* Compound **7c** was reacted according to General Procedure D, yield 86%, orange solid. ^1^H NMR (400 MHz, CD_3_OD, *δ*, ppm): 8.23 (d, *J* = 8.7 Hz, 2H), 7.79 (d, *J* = 2.1 Hz, 2H), 7.68 (dd, *J* = 8.7; 2.1 Hz, 2H), 7.49 (s, 2H). ^13^C NMR (100 MHz, CD_3_OD, *δ*, ppm): 157.88, 151.77, 136.35, 131.65, 131.37, 130.90, 129.34, 119.04, 113.89, 109.16. HRMS (ESI), found 385.1525 C_20_H_16_N_8_O_2_, [M + H]^+^, requires 385.1525.

*2,5-Bis(4-guanidino-2-(methyl carboxy)phenyl)furan (**8**).* Compound **8c** was reacted according to General Procedure D, yield 90%, orange solid. ^1^H NMR (400 MHz, CD_3_OD, *δ*, ppm): 7.80 (d, *J* = 8.4 Hz, 2H), 7.55 (d, *J* = 2.0 Hz, 2H), 7.51 (dd, *J* = 8.4; 2.0 Hz, 2H), 6.83 (s, 2H), 3.79 (s, 6H). ^13^C NMR (100 MHz, CD_3_OD, *δ*, ppm): 169.93, 157.86, 153.12, 135.96, 132.43, 130.39, 128.82, 128.45, 126.54, 112.08, 53.32. HRMS (ESI), found 451.1731 C_22_H_22_N_6_O_5_, [M + H]^+^, requires 451.1730.

*2,5-Bis(6-guanidino-2-methylpyridin-3-yl)furan (**9**).* Compound **9c** was reacted according to General Procedure D, yield 78%, tan solid. ^1^H NMR (400 MHz, CD_3_OD, *δ*, ppm): 8.29 (d, *J* = 8.5 Hz, 2H), 7.08 (d, *J* = 8.5 Hz, 2H), 7.00 (s, 2H), 2.85 (s, 6H). ^13^C NMR (100 MHz, CD_3_OD, *δ*, ppm): 157.07, 153.62, 151.95, 151.26, 138.71, 123.01, 112.94, 112.01, 24.79. HRMS (ESI), found 365.1837 C_18_H_20_N_8_O, [M + H]^+^, requires 365.1838.

*2,5-Bis(4-guanidinomethylphenyl)furan (**10**).* Compound **10** was reacted according to General Procedure D, yield 79%, pale yellow solid. ^1^H NMR (400 MHz, CD_3_OD, *δ*, ppm): 7.81 (d, *J* = 7.8 Hz, 4H), 7.41 (d, *J* = 7.8 Hz, 4H), 6.91 (s, 2H), 4.44 (s, 4H). ^13^C NMR (100 MHz, CD_3_OD, *δ*, ppm): 158.73, 154.34, 136.80, 131.74, 129.01, 128.85, 125.17, 125.00, 108.91, 45.75. HRMS (ESI), found 363.1930 C_20_H_22_N_6_O, [M + H]^+^, requires 363.1933. 

### 4.6. General Procedure E 

To a solution of 2,5-Bis(4-aminophenyl)furan derivatives (**1b**, **3b**–**5b**) (0.2 mmol) in 2.5 mL of ethanol was added 3,4-diethoxy-3-cyclobutene-1,2-dione (0.40 mmol) and zinc trifluoromethanesulfonate (0.08 mmol) in ethanol (2.5 mL). The solution was stirred at room temperature for 3 h. Then, the reaction mixture was evaporated under reduced pressure and the residue was subsequently dissolved in ethyl acetate and washed several times with aqueous ammonium chloride (1 M), and then with water. The organic layer was separated, dried over sodium sulfate, filtered, and triturated several times with pentane and diethyl ether. The resultant solid was dried under vacuum to give the desired compounds.

*2,5-Bis(2-chloro-4-((3,4-dioxo-2-(ethoxy)cyclobut-1-en-1-yl)amino)phenyl)furan (**11a**).* Compound **1b** was reacted according to General Procedure E, yield 52%, yellow solid. ^1^H NMR (400 MHz, CDCl_3_, *δ*, ppm): 8.24 (br s, 2H), 7.38 (m, 2H), 7.29 (br d, *J* = 8.0 Hz, 2H), 7.24 (d, *J* = 8.0 Hz, 2H), 7.14 (m, 2H), 4.92 (q, *J* = 6.8 Hz, 4H), 1.53 (t, *J* = 6.8 Hz, 6H).

*2,5-Bis(4-((3,4-dioxo-2-(ethoxy)cyclobut-1-en-1-yl)amino)phenyl)furan (**12a**).* Compound **3b** was reacted according to General Procedure E, yield 36%, red solid. ^1^H NMR (400 MHz, DMSO-*d*_6_, *δ*, ppm): 10.87 (br s, 2H), 7.80 (d, *J* = 7.8 Hz, 4H), 7.46 (d, *J* = 7.8 Hz, 4H), 7.01 (s, 2H), 4.78 (q, *J* = 6.9 Hz, 4H), 1.45 (t, *J* = 6.9 Hz, 6H).

*2,5-Bis(2-methyl-4-((3,4-dioxo-2-(ethoxy)cyclobut-1-en-1-yl)amino)phenyl)furan (**13a**).* Compound **4b** was reacted according to General Procedure E), yield 57%, yellow solid. ^1^H NMR (400 MHz, CDCl_3_, *δ*, ppm): 9.01 (br s, 2H), 7.22–7.16 (m, 6H), 6.95 (d, *J* = 7.1 Hz, 2H), 4.85 (q, *J* = 7.0 Hz, 4H), 2.33 (s, 6H), 1.47 (t, *J* = 7.0 Hz, 6H).

*2,5-Bis(4-((3,4-dioxo-2-(ethoxy)cyclobut-1-en-1-yl)amino)phenyl)furan (**14a**).* Compound **5b** was reacted according to General Procedure E, yield 75%, orange solid. ^1^H NMR (400 MHz, DMSO-*d*_6_, *δ*, ppm): 11.11 (br s, 2H), 7.96 (br s, 2H), 7.88 (d, *J* = 8.5 Hz, 2H), 7.72 (m, 2H), 6.92 (s, 2H), 4.81 (q, *J* = 7.0 Hz, 4H), 1.43 (t, *J* = 7.0 Hz, 6H).

### 4.7. General Procedure F 

To a solution of 2,5-Bis(4-((3,4-dioxo-2-(ethoxy)cyclobut-1-en-1-yl)amino)phenyl)furan derivatives (**11a**–**14a**) (0.1 mmol) in 1 mL of methanol was treated with ammonia (7 M in methanol, 200 µL, 1.45 mmol) and stirred at room temperature for 12 h. Then, the solvent was evaporated under reduced pressure and the residue was triturated several times with pentane and diethyl ether. The resultant solid was dried under vacuum to give the desired compounds.

*2,5-Bis(2-chloro-4-((3,4-dioxo-2-(amino)cyclobut-1-en-1-yl)amino)phenyl)furan (**11**).* Compound **11a** was reacted according to General Procedure F, yield 64%, yellow solid. ^1^H NMR (400 MHz, DMSO-*d*_6_, *δ*, ppm): 9.78 (s, 2H), 7.61 (br s, 2H), 7.27 (t, *J* = 8.1 Hz, 2H), 7.20 (m, 2H), 6.97 (d, *J* = 7.8 Hz, 2H). ^13^C NMR (100 MHz, DMSO-*d*_6_, *δ*, ppm): 184.76, 181.18, 171.37, 163.99, 140.73, 133.86, 130.95, 121.98, 117.62, 116.36. HRMS (ESI), found 509.0399 C_24_H_14_Cl_2_N_4_O_5_, [M + H]^+^, requires 509.0420. 

*2,5-Bis(4-((3,4-dioxo-2-(amino)cyclobut-1-en-1-yl)amino)phenyl)furan (**12**).* Compound **12a** was reacted according to General Procedure F, yield 56%, red solid. ^1^H NMR (400 MHz, DMSO-*d*_6_, *δ*, ppm): 10.05 (s, 2H), 7.77 (d, *J* = 8.4 Hz, 4H), 7.52 (d, *J* = 8.4 Hz, 4H), 6.94 (s, 2H). ^13^C NMR (100 MHz, DMSO-*d*_6_, *δ*, ppm): 184.44, 181.18, 171.25, 164.12, 152.11, 138.45, 124.53, 118.24, 109.55, 107.14. HRMS (ESI), found 441.1190 C_24_H_16_N_4_O_5_, [M + H]^+^, requires 440.1119. 

*2,5-Bis(2-methyl-4-((3,4-dioxo-2-(amino)cyclobut-1-en-1-yl)amino)phenyl)furan (**13**).* Compound **13a** was reacted according to General Procedure F, yield 52%, yellow solid. ^1^H NMR (400 MHz, DMSO-*d*_6_, *δ*, ppm): 9.62 (s, 2H), 7.20–7.16 (m, 6H), 6.79 (d, 7.0 Hz, 2H), 2.24 (s, 6H). ^13^C NMR (100 MHz, DMSO-*d*_6_, *δ*, ppm): 184.45, 181.20, 171.03, 164.58, 139.07, 138.70, 129.21, 123.29, 118.49, 115.15, 21.21. HRMS (ESI), found 469.1490 C_26_H_20_N_4_O_5_, [M + H]^+^, requires 469.1512. 

*2,5-Bis(2-trifluoromethyl-4-((3,4-dioxo-2-(amino)cyclobut-1-en-1-yl)amino)phenyl)furan (**14**).* Compound **14a** was reacted according to General Procedure F, yield 77%, orange solid. ^1^H NMR (400 MHz, DMSO-*d*_6_, *δ*, ppm): 10.14 (s, 2H), 8.02 (d, *J* = 1.8 Hz, 2H), 7.85 (d, *J* = 8.6 Hz, 2H), 7.70 (dd, *J* = 8.6; 1.8 Hz, 2H), 6.87 (s, 2H). ^13^C NMR (100 MHz, DMSO-*d*_6_, *δ*, ppm): 185.00, 181.29, 171.58, 163.58, 150.09, 139.48, 131.31, 126.04, 125.41, 121.60, 121.20, 115.93, 115.59. HRMS (ESI), found 577.0919 C_26_H_14_F_6_N_4_O_5_, [M + H]^+^, requires 577.0947. 

### 4.8. General Procedure G

To a solution of 2-furoyl chloride (2.8 mmol) in dichloromethane (15 mL) was added 5-nitroindoline or indoline (2.8 mmol) and triethylamine (1.6 mL), the mixture was stirred at room temperature for 3 h. The reaction mixture was monitored by GCMS, until the starting material was consumed. Then, the mixture was concentrated under reduced pressure and the crude product was purified by flash chromatography (25 g Sfär silica D cartridge, 15–50% B, R*_f_* = 8 column volumes) to give the desired compounds. 

*Furan-2-yl(5-nitroindolin-1-yl)methanone (2).* Synthesized from 5-nitroindoline, yield 50%, dark yellow solid. ^1^H NMR (400 MHz, DMSO-*d*_6_, *δ*, ppm): 8.22 (m, 1H), 8.17 (m, 2H), 8.02 (m, 1H), 7.38 (m, 1H), 6.76 (m, 1H), 4.55 (t, *J* = 8.3 Hz, 2H), 3.31 (m, 2H).

*Furan-2-yl(indolin-1-yl)methanone (**15**).* Synthesized from indoline, yield 59%, white solid. ^1^H NMR (400 MHz, DMSO-*d*_6_, *δ*, ppm): 8.04 (br d, *J* = 7.4 Hz, 1H), 7.92 (m, 1H), 7.26 (br d, *J* = 7.4 Hz, 1H), 7.23 (br d, *J* = 3.6 Hz, 1H), 7.17 (br t, *J* = 8.0 Hz, 1H), 7.03 (br t, *J* = 7.4 Hz, 1H), 6.69 (m, 1H), 4.38 (t, *J* = 8.2 Hz, 2H), 3.17 (t, *J* = 8.2 Hz, 2H).

### 4.9. General Procedure H 

In a sealed 20 mL microwave vial, the aryl bromide (1.1 mmol), furan-2-yl(indolin-1-yl)methanone (**2** or **15**) (0.9 mmol), potassium acetate (2.75 mmol), and palladium(II) acetate (0.02 mmol) were dissolved in dimethylacetamide (5 mL) and the resulting reaction mixture was evacuated and backfilled with nitrogen several times. The reaction was stirred at 150 °C for 20 h. Upon cooling to room temperature, the residue filtered through Celite and the Celite rinsed with ethyl acetate. The organic phase was washed with water and brine, dried over sodium sulfate, concentrated under reduced pressure. The crude product was purified by flash chromatography (25 g Sfär silica D cartridge, 25–100% B, R*_f_* = 12 column volumes) to give the desired compounds.

*(5-(2-chloro-4-nitrophenyl)furan-2-yl)(5-nitroindolin-1-yl)methanone (**16a**).* Compound **2** was reacted according to General Procedure H, 1-bromo-2-chloro-4-nitrobenzene was used as aryl bromide, yield 50%, red solid. ^1^H NMR (400 MHz, DMSO-*d*_6_, *δ*, ppm): 8.45 (m, 1H), 8.32 (m, 1H), 8.27 (m, 1H), 8.22 (m, 1H), 8.18 (m, 3H), 7.64 (dd, *J* = 3.8; 1.1 Hz, 1H), 7.59 (dd, *J* = 3.8; 1.1 Hz, 1H), 4.70 (t, *J* = 8.4 Hz, 2H), 3.33 (m, 2H).

*(5-nitroindolin-1-yl)(5-(4-nitrophenyl)furan-2-yl)methanone (**17a**).* Compound **2** was reacted according to General Procedure H, 1-bromo-4-nitrobenzene was used as the aryl bromide. Afforded the title compound as a yellow solid yield 43%. ^1^H NMR (400 MHz, DMSO-*d*_6_, *δ*, ppm): 8.34 (dd, *J* = 9.0; 1.0 Hz, 2H), 8.32 (m, 1H), 8.18 (m, 2H), 8.12 (dd, *J* = 9.0; 1.0 Hz, 2H), 7.56 (d, *J* = 1.0 Hz, 2H), 4.71 (t, *J* = 8.4 Hz, 2H), 3.35 (m, 2H).

*(5-(2-*trifluoromethyl*-4-nitrophenyl)furan-2-yl)(5-nitroindolin-1-yl)methanone (**18a**).* Compound **2** was reacted according to General Procedure H, 2-bromo-5-nitrobenzotrifluoride was used as the aryl bromide. Afforded the title compound as a brown solid, yield 31%. ^1^H NMR (400 MHz, DMSO-*d*_6_, *δ*, ppm): 8.59 (m, 2H), 8.24 (d, *J* = 6.28 Hz, 2H), 8.16 (m, 2H), 7.56 (d, *J* = 3.6 Hz, 1H), 7.30 (d, *J* = 3.6 Hz, 1H), 4.71 (t, *J* = 8.7 Hz, 2H), 3.30 (m, 2H).

*(5-(2-*chloro*-phenyl)furan-2-yl)(5-nitroindolin-1-yl)methanone (**19a**).* Compound **2** was reacted according to General Procedure H, 1-bromo-2-chlorobenzene was used as the aryl bromide. Afforded the title compound as a yellow solid, yield 38.3%. ^1^H NMR (400 MHz, CDCl_3_, *δ*, ppm): 8.35 (br d, *J* = 8.8 Hz, 1H), 8.16 (br d, *J* = 7.2 Hz, 1H), 8.10 (m, 1H), 7.82 (br d, *J* = 7.2 Hz, 1H), 7.50 (d, *J* = 7.6Hz, 1H), 7.43 (d, *J* = 3.5 Hz, 1H), 7.38 (t, *J* = 7.6 Hz, 1H), 7.32 (t, *J* = 7.6 Hz, 1H), 7.23 (d, *J* = 3.5 Hz, 1H), 4.72 (t, *J* = 8.4 Hz, 2H), 3.37 (t, *J* = 8.4 Hz, 1H).

*(5-(2-*chloro*-4-nitrophenyl)furan-2-yl)(5-indolin-1-yl)methanone (**20a**).* Compound **15** was reacted according to General Procedure H, 1-bromo-2-chloro-4-nitrobenzene was used as the aryl bromide. Afforded the title compound as a yellow solid, yield 36.5%. ^1^H NMR (400 MHz, DMSO-*d*_6_, *δ*, ppm): 8.38 (d, *J* = 2.3 Hz, 1H), 8.22 (m, 1H), 8.07 (br d, *J* = 8.8 Hz, 1H), 7.50 (d, *J* = 3.8 Hz, 1H), 7.35 (m, 1H), 7.26 (m, 3H), 7.10 (br t, *J* = 7.5 Hz, 1H), 4.55 (t, *J* = 8.1 Hz, 2H), 3.31 (t, *J* = 8.1 Hz, 2H).

### 4.10. General Procedure I 

To a solution of (5-nitroindolin-1-yl)(5-(4-nitrophenyl)furan-2-yl)methanone derivatives (**16a**–**18a**) or (5-nitroindolin-1-yl)(5-(phenyl)furan-2-yl)methanone (**19a**) or (indolin-1-yl)(5-(4-nitrophenyl)furan-2-yl)methanone (**20a**) (0.3 mmol) in tetrahydrofuran (3 mL) was added ethanol (3 mL) following was added 3 mL of ammonium chloride (0.3 M) and iron (1.75 mmol). After stirring at 60 °C for 4 h, the reaction was allowed to cool to room temperature and the heterogeneous mixture filtered through Celite and the Celite was rinsed with ethyl acetate. The solution was concentrated to half-volume, then diluted with ethyl acetate (20 mL) and washed with sodium hydroxide solution (1 M, 20 mL). The organic layer was separated, the aqueous phase was extracted with ethyl acetate (2×), the combined organic phases dried over sodium sulfate, filtered and the solvent evaporated. The crude products were used directly in the next step without further purification.

*(5-(4-amino-2-chlorophenyl)furan-2-yl)(5-aminoindolin-1-yl)methanone (**16b**).* Compound **16a** was reacted according to General Procedure I, yield 75%, yellow solid. ^1^H NMR (400 MHz, DMSO-*d*_6_, *δ*, ppm): 7.81 (br s, 1H), 7.53 (d, *J* = 8.7 Hz, 1H), 7.20 (dd, *J* = 3.6; 1.0 Hz, 1H), 6.90 (dd, *J* = 3.6; 1.0 Hz, 1H), 6.69 (dd, *J* = 2.2; 1.0 Hz, 1H), 6.62 (m, 1H), 6.49 (br s, 1H), 6.36 (m, 1H), 5.81 (br s, 2H), 4.94 (br s, 2H), 4.40 (t, *J* = 8.2 Hz, 2H), 3.08 (t, *J* = 8.2 Hz, 2H).

*(5-aminoindolin-1-yl)(5-(4-aminophenyl)furan-2-yl)methanone (**17b**).* Compound **17a** was reacted according to General Procedure I, yield 84%, fluffy yellow solid. ^1^H NMR (400 MHz, DMSO-*d*_6_, *δ*, ppm): 7.83 (br d, *J* = 7.5 Hz, 1H), 7.48 (d, *J* = 8.4 Hz, 2H), 7.18 (dd, *J* = 3.6; 0.5 Hz, 1H), 6.76 (dd, *J* = 3.6; 0.5 Hz, 1H), 6.63 (d, *J* = 8.4 Hz, 2H), 6.51 (d, *J* = 1.8 Hz, 1H), 6.38 (dd, *J* = 8.4; 1.8 Hz, 1H), 5.51 (br s, 2H), 4.94 (br s, 2H), 4.43 (t, *J* = 8.2 Hz, 2H), 3.10 (t, *J* = 8.2 Hz, 2H).

*(5-(4-amino-2-trifluoromethylphenyl)furan-2-yl)(5-aminoindolin-1-yl)methanone (**18b**).* Compound **18a** was reacted according to General Procedure I, yield 78.7%, tan solid. ^1^H NMR (400 MHz, DMSO-*d*_6_, *δ*, ppm): 8.09 (br s, 1H), 7.49 (d, *J* = 8.4 Hz, 1H), 7.25 (d, *J* = 4.1 Hz, 1H), 7.02 (d, *J* = 2.4 Hz, 1H), 6.84 (dd, *J* = 8.4; 2.4 Hz, 1H), 6.61 (d, *J* = 4.1 Hz, 1H), 6.56 (m, 2H), 4.69 (t, *J* = 8.2 Hz, 2H), 4.04 (br s, 2H), 3.13 (t, *J* = 8.2 Hz, 2H).

*(5-(2-chlorophenyl)furan-2-yl)(5-aminoindolin-1-yl)methanone (**19b**).* Compound **19a** was reacted according to General Procedure I, yield 90%, yellow solid. ^1^H NMR (400 MHz, CDCl_3_, *δ*, ppm): 8.11 (m, 1H), 7.84 (dd, *J* = 7.7; 1.5 Hz, 1H), 7.35 (td, *J* = 7.7; 1.5 Hz, 1H), 7.29 (d, *J* = 3.7; Hz, 1H), 7.28 (dd, *J* = 9.5; 1.6 Hz, 1H), 7.20 (d, *J* = 3.7; Hz, 1H), 6.57 (m, 2H), 4.53 (t, *J* = 8.3 Hz, 2H), 3.18 (t, *J* = 8.3 Hz, 2H).

*(5-(4-amino-2-chlorophenyl)furan-2-yl)(indolin-1-yl)methanone (**20b**).* Compound **20a** was reacted according to General Procedure I, yield 84%, yellow solid. ^1^H NMR (400 MHz, CDCl_3_, *δ*, ppm): 8.23 (m, 1H), 7.56 (dd, *J* = 8.5; 4.8 Hz, 1H), 7.20 (m, 3H), 7.03 (td, *J* = 7.4; 3.3 Hz, 1H), 6.96 (t, *J* = 3.9 Hz, 1H), 6.71 (dd, *J* = 4.8; 2.2 Hz, 2H), 6.59 (m, 1H), 4.47 (m, 2H), 3.21 (m, 2H).

### 4.11. General Procedure J 

To a solution of (5-aminoindolin-1-yl)(5-(4-aminophenyl)furan-2-yl)methanone derivatives (**16b**–**18b**) or (5-aminoindolin-1-yl)(5-(phenyl)furan-2-yl)methanone (**19b**) or (indolin-1-yl)(5-(4-aminophenyl)furan-2-yl)methanone (**20b**) (0.10 mmol) in dimethylformamide (5 mL) at 0 °C was added mercury (II) chloride (0.21 mmol), 1,3-bis(*tert*-butoxycarbonyl)-2-methyl-2-thiopseudourea (0.19 mmol) and triethylamine (0.48 mmol). The resulting mixture was stirred at 0 °C for 1 h and 18 h at room temperature. The reaction mixture was diluted with ethyl acetate, filtered through Celite and the Celite rinsed with ethyl acetate. The organic phase was washed with water (2×), and brine (2×), dried over sodium sulfate, filtered, and concentrated under reduced pressure. The crude product was purified by flash chromatography (10 g Sfär silica D cartridge, 10–45% B, R*_f_* = 10 column volumes) to give the desired compounds.

*(5-(2-chloro-4-N,N′-di*-(tert-butoxycarbonyl)*guanidinophenyl)furan-2-yl)(5-N,N′-di-(tert-butoxycarbonyl)guanidinoindolin-1-yl)methanone* (**16c**). Compound **16b** was reacted according to General Procedure J, yield 21%, yellow solid. ^1^H NMR (400 MHz, CDCl_3_, *δ*, ppm): 11.63 (m, 2H), 10.51 (d, *J* = 9.8 Hz, 1H), 10.34 (d, *J* = 9.8 Hz, 1H), 8.19 (br s, 1H), 7.80–7.67 (m, 4H), 7.31 (m, 1H), 7.20 (m, 1H), 7.15 (m, 1H), 4.57 (q, *J* = 8.6 Hz, 2H), 3.27 (q, *J* = 8.6 Hz, 2H), 1.51 (br s, 36H).

(5-(4-N,N′-di-(tert-butoxycarbonyl)guanidinophenyl)furan-2-yl)(5-N,N′-di-(tert-butoxycarbonyl)guanidinoindolin-1-yl)methanone (**17c**). Compound **17b** was reacted according to General Procedure J, yield 45%, yellow solid. ^1^H NMR (400 MHz, CDCl_3_, δ, ppm): 11.65 (m, 2H), 10.46 (s, 1H), 10.34 (s, 1H), 8.19 (br s, 1H), 7.74 (br s, 1H), 7.70 (d, J = 8.6 Hz, 2H), 7.66 (d, J = 8.6 Hz, 2H), 7.22 (m, 1H), 7.20 (br d, J = 8.6 Hz, 1H), 6.71 (m, 1H), 4.59 (t, J = 8.1 Hz, 2H), 3.28 (q, J = 8.1 Hz, 2H), 1.52 (br s, 36H).

*(5-(2-trifluoromethyl-4-N,N′-di*-*(tert-butoxycarbonyl)guanidinophenyl)furan-2-yl)(5-N,N′-di*-(tert-butoxycarbonyl)*guanidinoindolin-1-yl)methanone (**18c**).* Compound **18b** was reacted according to General Procedure J, yield 49%, tan solid. ^1^H NMR (400 MHz, CDCl_3_, *δ*, ppm): 11.64 (br s, 1H), 11.61 (br s, 1H), 10.61 (s, 1H), 10.34 (s, 1H), 8.09 (dd, *J* = 8.6; 2.1 Hz, 1H), 7.94 (d, *J* = 2.1 Hz, 1H), 7.74 (br s, 1H), 7.73 (d, *J* = 8.6 Hz, 1H), 7.32 (d, *J* = 3.6 Hz, 1H), 7.21 (dd, *J* = 8.6; 2.0 Hz, 1H), 6.77 (d, *J* = 3.6 Hz, 1H), 4.54 (q, *J* = 8.2 Hz, 2H), 3.25 (q, *J* = 8.2 Hz, 2H), 1.53 (br s, 36H).

*(5-(2-chlorophenyl)furan-2-yl)(5-N,N′-di-(tert-butoxycarbonyl)guanidinoindolin-1-yl)methanone* (**19c**)**.** Compound **19b** was reacted according to General Procedure J, yield 27%, yellow solid. ^1^H NMR (400 MHz, CDCl_3_, *δ*, ppm): 11.65 (s, 1H), 10.34 (s, 1H), 8.19 (br s, 1H), 7.83 (dd, *J* = 7.9; 1.4 Hz, 1H), 7.73 (br s, 1H), 7.46 (dd, *J* = 7.9; 1.3 Hz, 1H), 7.35 (td, *J* = 7.9; 1.3 Hz, 1H), 7.31 (d, *J* = 3.7 Hz, 1H), 7.27 (dd, *J* = 7.9; 1.7 Hz, 1H), 7.20 (m, 2H), 4.55 (t, *J* = 8.3 Hz, 2H), 3.27 (t, *J* = 8.3 Hz, 2H), 1.53 (s, 9H), 1.50 (s, 9H).

*(5-(2-chloro-4-N,N′-di-(tert-butoxycarbonyl)guanidinophenyl)furan-2-yl)(indolin-1-yl)methanone (**20c**).* Compound **20b** was reacted according to General Procedure I, yield 53%, yellow solid. ^1^H NMR (400 MHz, CDCl_3_, *δ*, ppm): 11.62 (s, 1H), 10.52 (s, 1H), 8.23 (br s, 1H), 7.81 (d, *J* = 4.8 Hz, 1H), 7.89 (d, *J* = 1.7 Hz, 1H), 7.71 (dd, *J* = 8.3; 2.1 Hz, 1H), 7.33 (d, *J* = 3.8 Hz, 1H), 7.23 (m, 1H), 7.16 (d, *J* = 3.8 Hz, 1H), 7.06 (m, 2H), 4.58 (t, *J* = 8.2 Hz, 2H), 3.28 (t, *J* = 8.2 Hz, 2H), 1.55 (s, 18H).

### 4.12. General Procedure K 

To a solution of (5-(4-*N*,*N*′-di-(tert-butoxycarbonyl)guanidinophenyl)furan-2-yl)(5-*N*,*N*′-di-(tert-butoxycarbonyl)guanidinoindolin-1-yl)methanone derivatives (**16c**–**18c**) or (5-phenyl)furan-2-yl)(5-*N*,*N*′-di-(tert-butoxycarbonyl)guanidinoindolin-1-yl)methanone (**19c**) or (5-(4-*N*,*N*′-di-(tert-butoxycarbonyl)guanidinophenyl)furan-2-yl)(inoindolin-1-yl)methanone (**20c**) (0.05 mmol) in dichloromethane (1 mL) was added HCl (4 M in dioxane, 2 mL). The solution was stirred at room temperature for 16 h. The mixture was concentrated under reduced pressure and the residue was purified by reverse phase flash chromatography (6 g Sfär C18 D cartridge, 0–50% B, R*_f_* = 10 column volumes) to give the desired compounds.

*(5-(2-chloro-4-guanidinophenyl)furan-2-yl)(5-guanidinoindolin-1-yl)methanone (**16**).* Compound **16c** was reacted according to General Procedure K, yield 86%, yellow solid. ^1^H NMR (400 MHz, CD_3_OD, *δ*, ppm): 8.26 (br d, *J* = 8.5 Hz, 1H), 8.09 (d, *J* = 8.5 Hz, 1H), 7.51 (d, *J* = 2.0 Hz, 1H), 7.43 (d, *J* = 3.8 Hz, 1H), 7.39 (dd, *J* = 8.5; 2.0 Hz, 1H), 7.36 (d, *J* = 3.8 Hz, 1H), 7.24 (br s, 1H), 7.15 (dd, *J* = 8.5; 2.0 Hz, 1H), 4.64 (t, *J* = 8.3 Hz, 2H), 3.35 (t, *J* = 8.3 Hz, 2H). ^13^C NMR (100 MHz, CD_3_OD, *δ*, ppm): 158.93, 158.27, 157.79, 153.38, 148.10, 143.80, 137.63, 136.03, 132.88, 132.18, 131.07, 127.81, 127.78, 125.92, 124.66, 123.69, 120.54, 119.62, 114.99, 51.31, 29.50. HRMS (ESI), found 438.1441 C_21_H_20_ClN_7_O_2_, [M + H]^+^, requires 438.1445.

*(5-(4-guanidinophenyl)furan-2-yl)(5-guanidinoindolin-1-yl)methanone (**17**).* Compound **17c** was reacted according to General Procedure K, yield 35%, pale yellow solid. ^1^H NMR (400 MHz, CD_3_OD, *δ*, ppm): 8.25 (d, *J* = 8.6 Hz, 1H), 7.96–7.93 (m, 2H), 7.39 (m, 3H), 7.24 (m, 1H), 7.14 (dd, *J* = 8.5; 2.0 Hz, 1H), 7.36 (d, *J* = 3.8 Hz, 1H), 7.24 (br s, 1H), 7.15 (dd, *J* = 8.5; 2.3 Hz, 1H), 7.08 (d, *J* = 3.8 Hz, 1H), 4.64 (t, *J* = 8.2 Hz, 2H), 3.35 (t, *J* = 8.2 Hz, 2H). ^13^C NMR (100 MHz, CD_3_OD, *δ*, ppm): 159.08, 158.27, 157.89, 157.03, 148.02, 143.87, 136.71, 135.98, 132.09, 129.76, 127.26, 126.56, 125.89, 123.67, 121.10, 119.58, 108.96, 51.24, 29.50. HRMS (ESI), found 404.1833 C_21_H_21_N_7_O_2_, [M + H]^+^, requires 404.1835.

*(5-(2-trifluoromethy-4-guanidinophenyl)furan-2-yl)(5-guanidinoindolin-1-yl)methanone (**18**).* Compound **18c** was reacted according to General Procedure K, yield 80%, tan solid. ^1^H NMR (400 MHz, CD_3_OD, *δ*, ppm): 8.26 (br d, *J* = 8.6 Hz, 1H), 8.02 (d, *J* = 8.5 Hz, 1H), 7.79 (d, *J* = 2.1 Hz, 1H), 7.70 (dd, *J* = 2.1; 8.5 Hz, 1H), 7.44 (d, *J* = 3.6 Hz, 1H), 7.25 (d, *J* = 2.0 Hz, 1H), 7.16 (dd, *J* = 2.1; 8.6 Hz, 1H), 7.00 (d, *J* = 3.6 Hz, 1H), 4.61 (t, *J* = 8.3 Hz, 2H), 3.31 (m, 2H). ^13^C NMR (100 MHz, CD_3_OD, *δ*, ppm): 158.90, 158.29, 157.90, 153.66, 149.33, 143.78, 137.84, 136.05, 133.87, 132.19, 127.84, 129.31, 125.95, 124.13 (d, *J* = 5.38 Hz), 123.72, 123.39 120.49, 119.62, 113.62 (q, *J* = 6.3 Hz), 51.12, 29.43. HRMS (ESI), found 472.1708 C_22_H_20_F_3_N_7_O_2_, [M + H]^+^, requires 472.1709.

*(5-(2-chlorophenyl)furan-2-yl)(5-guanidinoindolin-1-yl)methanone (**19**).* Compound **19c** was reacted according to General Procedure K, yield 81%, yellow solid. ^1^H NMR (400 MHz, CD_3_OD, *δ*, ppm): 8.24 (br d, *J* = 8.5 Hz, 1H), 7.96 (dd, *J* = 7.9; 1.5 Hz, 1H), 7.54 (dd, *J* = 7.9; 1.5; Hz, 1H), 7.45 (dt, *J* = 1.5; 7.6 Hz, 1H), 7.39 (d, *J* = 3.8 Hz, 1H), 7.37 (dt, *J* = 1.6; 7.6 Hz, 1H), 7.30 (d, *J* = 3.8 Hz, 1H), 7.23 (br d, *J* = 1.9, 1H), 7.13 (dd, *J* = 1.9; 8.5 1H), 4.63 (t, *J* = 8.3 Hz, 2H), 3.34 (t, *J* = 8.3 Hz, 2H). ^13^C NMR (100 MHz, CD_3_OD, *δ*, ppm): 158.88, 158.24, 154.19, 147.94, 143.81, 135.95, 132.05, 132.03, 130.99, 129.87, 129.17, 128.55, 125.83, 123.60, 120.61, 119.55, 113.68, 51.18, 29.44. HRMS (ESI), found 381.1118 C_20_H_17_ClN_4_O_2_, [M + H]^+^, requires 381.1118.

*(5-(2-chloro-4-guanidinophenyl)furan-2-yl)(indolin-1-yl)methanone (**20**).* Compound **20c** was reacted according to General Procedure K, yield 58%, yellow solid. ^1^H NMR (400 MHz, CD_3_OD, *δ*, ppm): 10.27 (br s, 1H), 8.2009 (br d, *J* = 8.6 Hz, 1H), 7.93 (dd94 (d, *J* = 7.8; 1.6 Hz, 1H), 7.51 (dd74 (br s, 2H), 7.47 (d, *J* = 7.8; 1.3 Hz, 1H), 7.41 (td, *J* = 7.8; 1.32.1 Hz, 1H), 7.35 (m, 2H), 7.39 (d, *J* = 3.7 Hz, 1H), 7.34 (dd, *J* = 8.6; 2.1 Hz, 1H), 7.29 (br s, 1H), 7.26 (d, *J* = 3.87 Hz, 1H), 7.19 (m, 1H), 7.10 (dd, *J* = 8.6; 2.2 Hz04 (m, 1H), 4.52 (t, *J* = 8.2 Hz, 2H), 3.22 (t, *J* = 8.2 Hz, 2H). ^13^C NMR (100 MHz, DMSO-*d*_6_, *δ*, ppm): 165.72, 165.22, 160.34, 156.63, 152.50, 146.37, 141.79, 139.86, 139.04, 136.43, 134.99, 134.38, 133.58, 132.43, 128.28, 126.37, 120.54, 121.84, 58.50, 37.63. HRMS (ESI), found 381.1118 C_20_H_17_ClN_4_O_2_, [M + H]^+^, requires 381.1118.

### 4.13. Antibacterial Activity Assays

All compounds were dissolved at 50 mM in 100% DMSO and stored at −20 °C until analysis. The antibacterial activity of all compounds was determined against non-pathogenic Gram-positive bacteria *B. subtilis* (NBRC/ATCC #111470), and four Gram-negative bacteria; *E. coli* MG1655 (CGSC #6300), *P. putida* (NBRC/ATCC #100650), *P. carotovorum* (NBRC/ATCC #3380) and *P. caledonica* (NBRC/ATCC #102488). The bacteria were cultured as previously described by Mueller and Hinton [38] and Doyle [39]. For in vitro determination of antibacterial activity, a culture of bacterial cells was grown to OD 600_nm_ = 0.5. The bacterial culture was diluted 10× with pre-warmed fresh medium and aliquoted into a 384-well plate before adding compounds. The starting concentration of the compound was either 300 or 1000 µM, and following dilution in two-fold intervals, the plate was incubated at 37 °C for 18 h without agitation. To measure cell viability, we used the resazurin-based assay as described previously [40]. To each well, 12 μL of 10× AlamarBlue solution (resazurin solution, ThermoFisher, Waltham, MA, USA) was added, and a 384-well plate incubated at 37 °C for 1 h. The 384-well plates enabled us to use small final volumes of 20 µL conserving compound stock and slowing down evaporation due to the depth of the wells. Fluorescence was measured using a POLARstar Omega microplate-reader (BMG Labtech, Offenburg, Germany) with excitation filter set to 544 nm and emission filter to 590 nm. Cells exposed to only the equivalent concentration of DMSO were used as negative control. Bleed-through of fluorescence from resorufin between wells in the microtiter plate fluorescence reader, was measured and found to be <1% between adjacent wells. The 384-well plates were used to avoid this fluorescence bleed-through, achieved by skipping a well in-between bacterial cultures and compound dilutions. To check for quenching of fluorescence by any of the investigated compounds, grown bacterial cultures were mixed after 1 h incubation with resazurin and the compound of interest at the highest concentration to be assayed, and the measured fluorescence compared with samples without compound added. All tests of compound activity were performed in three independent replicates for more accurate determination of the half (EC_50_) and 90% maximal effective concentration (EC_90_). 

The antibacterial activities of eight compounds (**1**, **3**, **4**, **6**, **8**, **10**, **16,** and **17**) were tested against 10 different isolates of Gram-negative bacteria from human and other sources, *two E. coli* (CCUG #67180 and CCUG #17620/ATCC #25922, control strain), two *K. pneumoniae* (CCUG #58547 and CCUG #225T), two *P. aeruginosa* (CCUG #17619 and CCUG #59347), two *A. baumannii* (CCUG #57035 and CCUG #57250), *E. cloacae* (CCUG #6323T), and *E. hormaechei* (CCUG #58962) in a 96-well plate format. The compounds to test were 3-fold diluted in six steps in cation-adjusted Mueller-Hinton (ca-MH) broth to final concentrations spanning from 30 to 0.12 µM, i.e., a 1667–416,667 times dilution from the 50 mM stocks. Bacterial mass from the ten isolates grown overnight on horse blood or Mueller–Hinton agar plates was suspended in ca-MH and adjusted to a final inoculum cell density of ~5 × 10^5^ CFU/mL. For all assays, Biolog redox dye A diluted 100× from the stock solution was used for measurements of all ten isolates (up to a total volume of 120 µL per well). All plates included one well per isolate with only inoculum and dye but no test compound (i.e., positive control) and one well per isolate with inoculum, dye and DMSO diluted 1667×, corresponding to the DMSO concentration in the wells with the highest concentration of test compound. All measurements were performed using the Omnilog microplate reader (Biolog, Hayward, CA, USA) where the 96-well plates were read at 15 min intervals for a total of 24 h at 37 °C. Each compound was run in triplicate on three independent assay plates for more accurate determination of the half (EC_50_) and 90% maximal effective concentration (EC_90_).

Non-linear regression dose–response inhibition following a log (agonist) vs. response–Find ECanything was performed using GraphPad Prism version 9.2.0 for Windows, GraphPad Software, San Diego, CA, USA, www.graphpad.com (accessed on 20 July 2022).

### 4.14. Cytotoxicity Assays

The cytotoxicity levels of all compounds were evaluated against human Michigan Cancer Foundation-7 (MCF-7) and Hepatoma G2 cell line (HepG2) cell lines. MCF-7 is an extensively characterized breast cancer cell line isolated in 1970 [41], while HepG2 cells were derived from a hepatoma in 1975 [42]. Both cell lines grow robustly during in vitro culture and have been widely used for cytotoxicity testing. Cells of hepatic origin, such as HepG2, are of particular relevance for toxicity studies as many drugs accumulate in the liver during metabolic conversion [41,42]. Cells were grown in Dulbecco’s Modified Eagle Medium supplemented with 10% fetal calf serum and kept in exponential growth, as previously reported [43]. Before the assay, cells were reseeded into 96-well microtiter plates at a density allowing continued exponential growth and let to settle for 24 h. The compounds were added from a stock solution in DMSO, for a final concentration of 0.3% *v*/*v* of the solvent in the culture medium. After 24 h of incubation in presence of the compound, cell viability was assayed using PrestoBlue Cell Viability Reagent (resazurin-based solution, ThermoFisher, Waltham, MA, USA) according to the manufacturer’s instructions. A POLARstar Omega microplate-reader (BMG Labtech, Offenburg, Germany) was used to measure resorufin fluorescence at 544 nm excitation/590 nm emissions. Each assay contained a DMSO control at the equivalent starting concentration, positive control (uninhibited cell growth) and negative control (cell medium only). Survival was expressed as percentage of the solvent-only control. EC_50_ values for each compound were calculated from three independent replicate experiments using 2-fold dilution intervals. Non-linear regression dose–response inhibition following a log(agonist) vs. response–Find ECanything was performed using GraphPad Prism version 9.2.0 for Windows, GraphPad Software, San Diego, California USA, www.graphpad.com. The Selectivity Index (SI) for each compound and bacterial strain was calculated as the ratio between the mean of the EC_50_ values for the two human cell lines and the EC_50_ for the bacterial strain in question ((EC_50_ MCF-7 + EC_50_ HepG2)/2)/EC_50_ bacterial strain). The higher the value, the more selective is the compound against the different bacterial strain.

### 4.15. High-Resolution Microbial Phenomics (Scan-o-Matic) Assays

*E. coli* colonies were deposited as initially isogenic populations at initial population sizes of ~100,000 cells, with 1536 colonies deposited in systematic colony arrays on each plate on top of a solid matrix composed of LB medium supplemented with a sublethal concentration (60 µM) of five different compounds, **1**, **4**, **6**, **10**, and **16**, and a known antimicrobial, cefotaxime (CTX; 2 µg/mL), using automated pinning by robot. The compound concentrations were empirically chosen to strongly but not completely inhibit colony growth on agar of the more sensitive strains in the collections, in order to enable quantitation of the difference in growth yield between more and less resistant strains [44,45]. Of these colonies, 384 were identical controls used to correct for spatial bias between and within plates. For CTX and **4**, **10**, and **16**, each lineage was cultivated as six biological replicates on different plates. For **1**, nine biological replicates were done. Population expansion for each colony was followed by measuring cell numbers at 10 min intervals using the Scan-o-Matic framework, version 2.0 [44] with an *E. coli* calibration curve [45]. From each colony growth curve, the total cell yield after 8 h was extracted (growth yield). Experiments included automated transmissive scanning and signal calibration in 10 min intervals, as described [44]. The absolute population yields were log(2) transformed and normalized to the corresponding measures of adjacent controls (fourth position *E. coli* CCUG #17620/ATCC #25922 strain) on each plate, while data for missing or mis-quantified colonies were discarded. The relative growth yield of each strain (total n = 164) of the screened ECOR (n = 72) and ESBL (n = 92) libraries on the tested compounds was normalized to their corresponding growth without compound and the resulting ratios were clustered and visualized using heatmaps constructed using the R package *ComplexHeatmap* v. 2.8.0 [27].

## 5. Conclusions

There is a paucity of antibiotics effective against Gram-negative bacteria among which multiple resistance has spread. We have identified a group of small molecules that have demonstrated promising activity against antibiotics-resistant Gram-negative pathogens of the ESKAPE group and *E. coli*, which pose a major clinical concern. As with all new antibacterial agents, resistance to these molecules is likely to eventually occur among bacterial populations. Even if bacterial gene products dedicated to inactivate these new compounds do not exist, resistance can develop through e.g., increased efflux, blocked uptake, and increased target expression. Further, the efficacy against persister cells or biofilms has not been tested. 

The molecular target of these diaryl compounds remains to be established, however they represent a promising resource for further development of antibiotics, or as potentiators of other antibiotics. Our modifications of the lead molecule for reduced cytotoxicity and greater activity against a broader set of bacterial species represent important steps in this direction. 

## Figures and Tables

**Figure 1 antibiotics-11-01115-f001:**
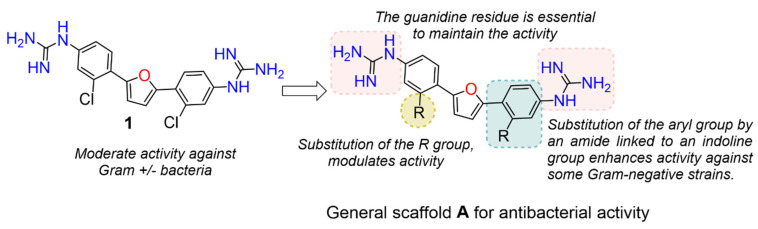
Structure of 2,5-bis(2-chloro-4-guanidinophenyl)furan **1** and structure–activity relationship (SAR) evaluation plan.

**Figure 2 antibiotics-11-01115-f002:**
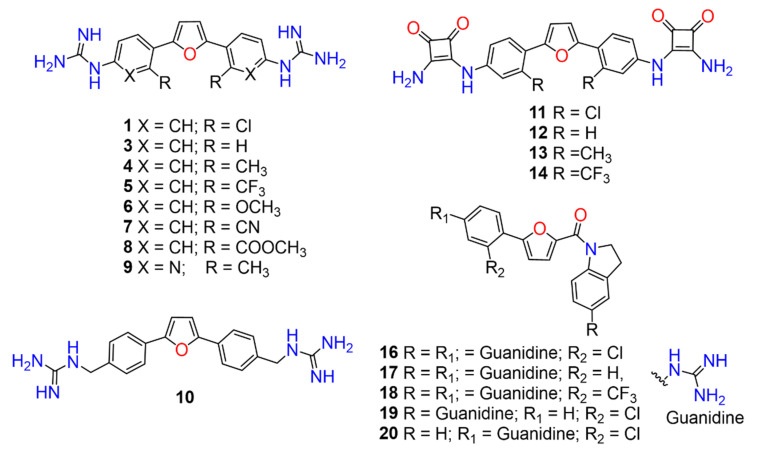
Chemical structures of the synthesized guanidium-arylfurans (**1**, **3**–**10**, **16**–**20**) and squaramides-arylfurans (**11**–**14**).

**Figure 3 antibiotics-11-01115-f003:**
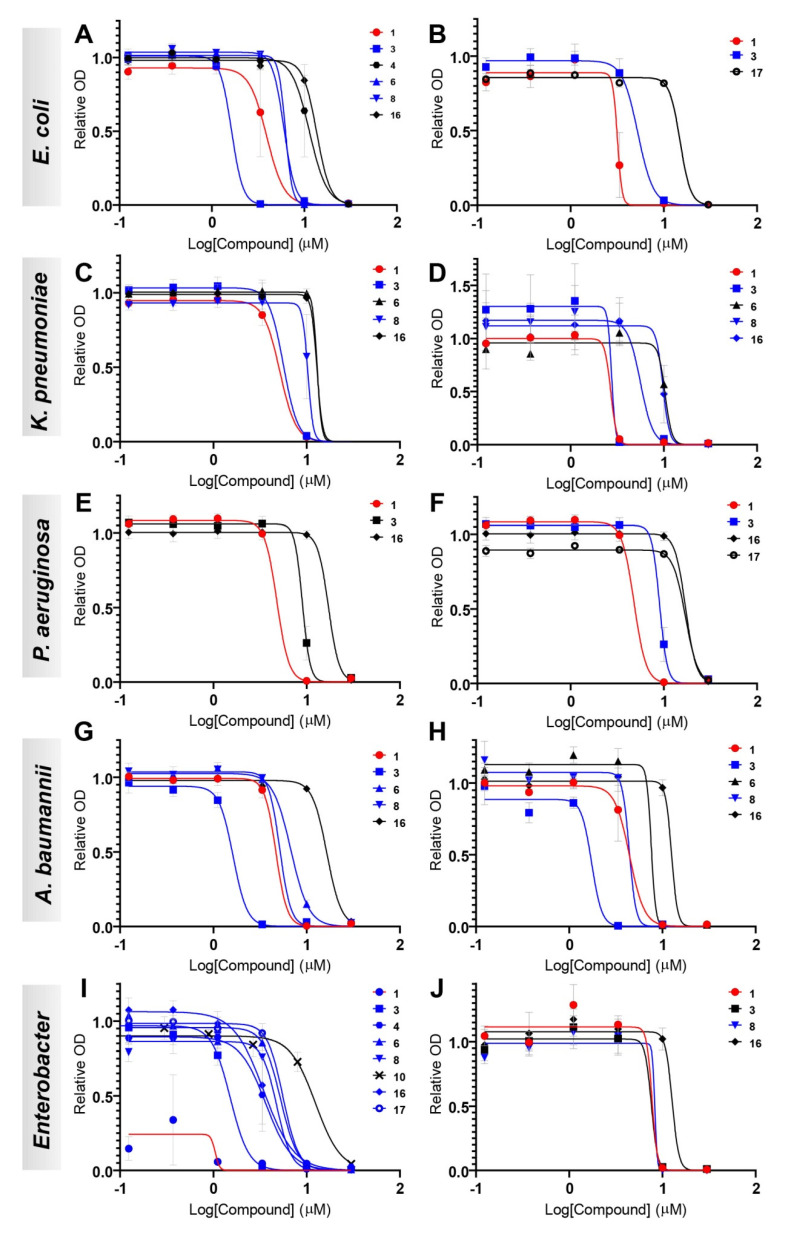
Antibacterial activity dose–response curves of compounds showing the most promising antimicrobial effects against the 10 Gram-negative bacteria, *E. coli* CCUG #17620/ATCC #25922 (control strain) (**A**) and CCUG #67180 (**B**), *K. pneumoniae* CCUG #58547 (**C**) and CCUG #225T (**D**), *P. aeruginosa* CCUG #17619 (**E**) and CCUG #59347 (**F**), *A. baumannii* CCUG #57035 (**G**) and CCUG #57250 (**H**), *E. cloacae* CCUG #6323T (**I**) and *E. hormaechei* CCUG #58962 (**J**). Compounds with the highest selectivity indices (SI ≥ 20×) are shown in blue and the lead compound in red. EC_50_, EC_90_ and SI values (in µM) can be found in Table 2 and Appendix A.

**Figure 4 antibiotics-11-01115-f004:**
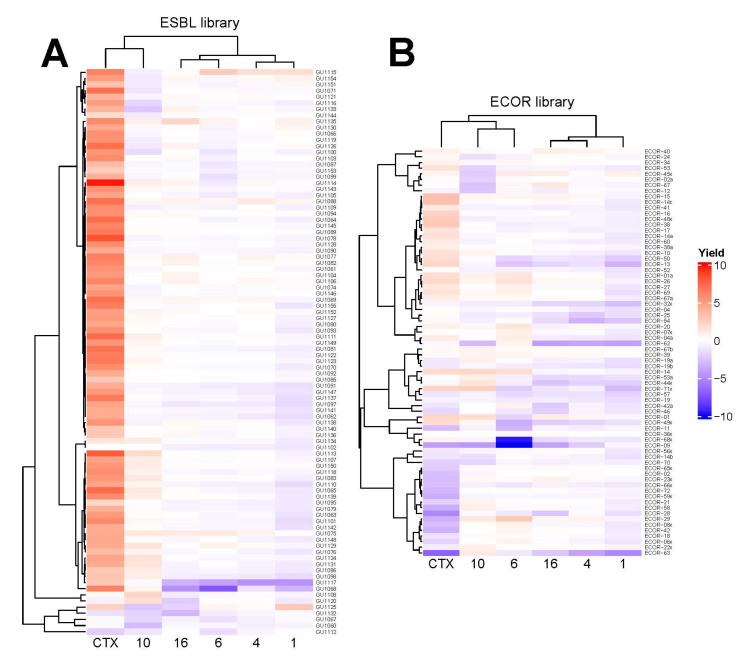
High-resolution microbial phenomics profiling of synthesized compounds **1**, **4**, **6**, **10**, **16,** and known antibiotic CTX against two *E. coli* libraries (ECOR and ESBL). Heatmap clustering of the growth yield upon exposure to the compounds relative to our reference strain (*E. coli* CCUG #17620/ATCC #25922) normalized for growth without added compound for the 92 strains of the ESBL library (**A**) and for the 72 strains of the ECOR library (**B**), using complete linkage hierarchical clustering method and Pearson’s distance measure method for computing distance between rows and columns. The values are expressed on a log(2) scale where positive and negative values indicate better (i.e., more resistant) and worse (i.e., more sensitive) yield compared to the control. The clustering and construction of the heatmaps was performed using the R package *ComplexHeatmap* v. 2.8.0 [27].

**Table 1 antibiotics-11-01115-t001:** Cytotoxicity of each compound against MCF-7 and HepG2 human cell lines.

		MCF-7	HEPG2
Structure	Code	EC_50_	EC_50_
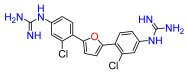	**1**	34.4	43.4
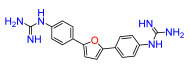	**3**	64.5	184.7
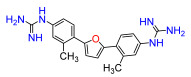	**4**	>100	51.3
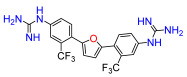	**5**	23.2	69.0
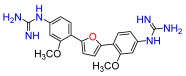	**6**	74.7	213.2
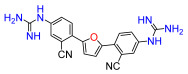	**7**	46.0	78.9
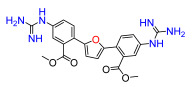	**8**	81.6	404.8
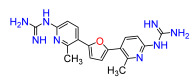	**9**	>1000	>100
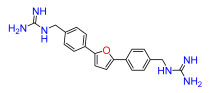	**10**	48.6	137.1
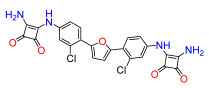	**11**	507.5	183.3
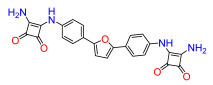	**12**	>1000	>1000
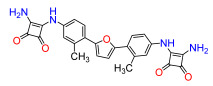	**13**	>1000	137.4
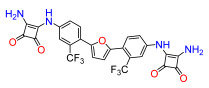	**14**	172.6	185.4
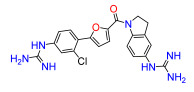	**16**	177.1	240.7
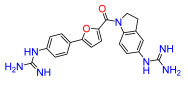	**17**	149.1	239.4
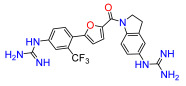	**18**	123.1	508.3
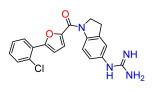	**19**	26.4	24.8
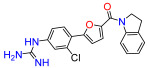	**20**	37.8	28.5

Half maximal effective concentration (EC_50_) in µM for each compound. Values >1000 and >100 µM represent the maximum compound concentration tested in the cytotoxicity assays, without observing 50% inhibition. MCF-7: Michigan Cancer Foundation-7 cell line; HepG2: Hepatoma G2 cell line. Cytotoxicity dose–response curves for all compounds are shown in Appendix A.

**Table 2 antibiotics-11-01115-t002:** Effective concentrations of **1**, **3**, **4**, **6**, **8**, **10**, **16**, **17** and CTX against the Gram-negative set of ESKAPE and *E. coli* isolates with different antibiotic susceptibility profiles.

**Code**	***K. pneumoniae*** (CCUG #58547)		***K. pneumoniae*** (CCUG #225T)		***P. aeruginosa*** (CCUG #17619)		***P. aeruginosa*** (CCUG #59347)		***A. baumannii*** (CCUG #57035)	
	EC_50_	EC_90_	EC_50_	EC_90_	EC_50_	EC_90_	EC_50_	EC_90_	EC_50_	EC_90_
**1**	5.2	8.0	2.7	3.2	4.8	6.7	5.3	7.4	4.7	6.3
**3**	5.6	8.3	2.8	3.1	9.0	11.1	5.2	8.1	1.6	2.4
**4**	17.1	22.4	15.3	22.7	17.9	25.4	66.8	77.0	16.0	23.5
**6**	13.0	13.7	10.3	12.7	24.6	28.7	54.8	65.3	6.8	11.0
**8**	10.3	13.3	5.6	7.9	NE	NE	NE	NE	5.2	7.0
**10**	NE	NE	NE	NE	NE	NE	NE	NE	NE	NE
**16**	12.9	15.3	9.8	12.2	17.1	22.9	18.1	24.0	16.5	24.4
**17**	NE	NE	NE	NE	22.9	27.0	17.0	23.9	148.1	>200
**CTX**	324.1	478.1	0.2	0.3	13.0	18.4	1510	1684	430.0	680.6
**Code**	***A. baumannii*** (CCUG #57250)		***E. cloacae*** (CCUG #6323T)		***E. hormaechei*** (CCUG #58962)		***E. coli*** (CCUG #17620/ ATCC #25922)		***E. coli*** (CCUG #67180)	
	EC_50_	EC_90_	EC_50_	EC_90_	EC_50_	EC_90_	EC_50_	EC_90_	EC_50_	EC_90_
**1**	4.4	6.6	1.0	1.2	7.7	9.1	3.9	6.3	3.2	3.8
**3**	1.7	2.3	1.5	2.5	7.6	9.1	1.6	2.2	5.3	8.0
**4**	15.8	22.5	3.7	7.4	21.9	25.9	11.3	18.3	23.1	25.3
**6**	7.6	8.8	5.3	8.4	15.9	22.9	6.2	7.5	63.4	74.4
**8**	4.4	5.1	4.8	7.1	8.3	8.7	6.0	8.2	53.4	67.5
**10**	NE	NE	12.4	24.7	NE	NE	150.7	179.2	188.5	>200
**16**	12.5	12.5	3.5	7.9	12.7	14.7	13.6	19.7	23.0	27.1
**17**	61.8	90.3	5.6	8.6	NE	NE	16.4	24.0	14.2	19.9
**CTX**	23.4	48.0	2.1	4.5	78.4	158.3	0.2	0.3	161.2	183.1

Half maximal effective concentration (EC_50_) and 90% maximal effective concentration (EC_90_) are expressed in µM for each compound. NE, No Effective concentration observed. CCUG: Culture Collection University of Gothenburg. Resistance levels of these isolates to established antibiotics are shown in Appendix A. Dose–response curves for all compounds are shown in Appendix A.

**Table 3 antibiotics-11-01115-t003:** Selectivity indices of each compound against the Gram-negative set of ESKAPE isolates and *E. coli*.

**Code**	***E. coli*** (CCUG #17620/ ATCC #25922)	***E. coli*** (CCUG #67180)	***K. pneumoniae*** (CCUG #58547)	***K. pneumoniae*** (CCUG #225T)	***P. aeruginosa*** (CCUG #17619)
**1**	9.9	12.2	7.5	14.4	8.1
**3**	76.6	23.7	22.1	45.2	13.8
**4**	6.7	3.3	4.4	4.9	4.2
**6**	23.1	2.3	11.1	14.0	5.9
**8**	40.5	4.6	23.6	43.3	ND
**10**	0.6	0.5	ND	ND	ND
**16**	15.4	9.1	16.2	21.4	12.2
**17**	11.8	13.7	ND	ND	8.5
**Code**	***P. aeruginosa*** (CCUG #59347)	***A. baumannii*** (CCUG #57035)	***A. baumannii*** (CCUG #57250)	***E. cloacae*** (CCUG #6323T)	***E. hormaechei*** (CCUG #58962)
**1**	7.3	8.3	8.7	37.2	5.0
**3**	24.1	76.5	71.5	83.1	16.4
**4**	1.1	4.7	4.8	20.7	3.4
**6**	2.6	21.3	19.0	27.4	9.0
**8**	ND	47.2	55.3	50.8	29.4
**10**	ND	ND	ND	7.5	ND
**16**	11.5	12.7	16.7	60.1	16.4
**17**	11.4	1.3	3.1	34.8	ND

Average selectivity indices (SI) for a particular compound are calculated as the ratio of the mean of the cytotoxicity EC_50_ value verified for each human cell line (MCF-7 and HepG2) (Table 1) over each bacterial strains’ EC_50_ values (Table 2); SI = ((EC_50_ MCF-7 + EC_50_ HepG2)/2)/EC_50_ bacterial strain). The higher the value, the more selective is the compound against the different bacterial strain. ND, Not Determined. CCUG: Culture Collection University of Gothenburg.

## Data Availability

All data is found in the Appendix A provided with this publication.

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
