# Peer review of "Development of Dicationic Bisguanidine-Arylfuran Derivatives as Potent Agents against Gram-Negative Bacteria"

_antibiotics, 2022, doi:10.3390/antibiotics11081115_

Round 1
Reviewer 1 Report
The manuscript presented for review is a report about the dicationic bisguanidine-arylfuran derivatives as potent agents against Gram-negative bacteria. The spread of antibiotic resistance among bacteria has become one of the major health problems worldwide therefore, I believe that the presented research are important.
The methodology used by the authors is appropriate. The results are presented in a very precise way. Overall, this is a clear and well organized manuscript and it is necessary to do only minor revision in the text before acceptation:
Please remove the period from the title
Line 14-15: Please correct
Line 31: Escherichia coli à italic
Line 44: targetà targets
Figure 2 and 3 are not clear, please enlarge the structural formulas and charts if possible
Table 1 also have not clear structural formulas
Line 121: the color of font is diffrent
Line 544 à font
Line 964, 967, 972 à too large font
Author Response
The manuscript presented for review is a report about the dicationic bisguanidine-arylfuran derivatives as potent agents against Gram-negative bacteria. The spread of antibiotic resistance among bacteria has become one of the major health problems worldwide therefore, I believe that the presented research are important.
The methodology used by the authors is appropriate. The results are presented in a very precise way. Overall, this is a clear and well organized manuscript and it is necessary to do only minor revision in the text before acceptation:
- Please remove the period from the title
Response: This period was not there in the original file provided by us, but added in the MDPI template
- Line 14-15: Please correct
Response: The misalignment of the names of the corresponding authors was not there in the file provided by us. It must have resulted by insertion into the MDPI template
- Line 31: Escherichia coli à italic
Response: Again, the italics were there in the original file, but were apparently removed by MDPI staff when inserting into the template.
- Line 44: targetà targets
Response: This should actually be “target”, as it refers to the few products (plural) that target Gram-negative bacteria. Text has not been changed.
- Figure 2 and 3 are not clear, please enlarge the structural formulas and charts if possible
Response: Figure 2 has been modified and the structural formulas enlarged. In Figure 3, the panels with graphs have also been enlarged to improve visibility.
- Table 1 also have not clear structural formulas
Response: Table 1 has been modified, and the structural formulas enlarged.
- Line 121: the color of font is different
Response: In line 121, which reads “tidomimetics [20]. The 1,2-diaminocyclobutene-3,4-dione (squaryldiamide) derivatives”, we see no different font colors in this line, neither in the original submitted file nor in the MDPI-formatted pdf.
- Line 544 à font
Response: This was an error in our original file. Now, the same font as in the rest of the text body has been inserted here. The same error also occurred at four more instances in the lines above this, they have also been corrected.
- Line 964, 967, 972 à too large font
Response: We used the same size font for these headings (Acknowledgements, Funding information and Conflicts of interest) as for all other headings in the originally submitted file. The change in font size must have taken place during staff handling of the document.
Reviewer 2 Report
Should be published in Antibiotics after minor revision
In their manuscript entitled “Development of dicationic bisguanidine-arylfuran derivatives as potent agents against Gram-negative bacteria”, Catarina Bourgard et al. report the design and synthesis of a series of dicationic derivatives. These compounds were used to examine their antibacterial activity against antibiotics-resistant Gram-negative pathogens of the ESKAPE group and E. coli, cytotoxicity against two human cell lines, and possible mechanism of action.
General appreciation
In my opinion, this study is worth of interest, reporting promising new dicationic bisguanidine-arylfuran derivatives and their thorough characterizations considering a panel of relevant bacterial pathogens. Adequate well-detailed experimental procedures were employed, which allowed obtaining many original data reported in a comprehensive, interesting to read manuscript. I have a list of some only minor suggestions/comments, which is given below. In particular, I think some explanations are missing in multiple instances and data presentation could be improved. Accordingly, I recommend a minor revision before publication of this manuscript in Antibiotics.
Specific comments
Purity of compounds should be discussed in the text.
More details about the cell lines, their origin and the reasons for their use to determine cytotoxicity, should be given.
Authors state about selectivity or therapeutic index. The calculation used is not explicitly given.
The manuscript contains some typo errors. A thorough reading should allow to correct these. For example in Figure 1: “against”, “Essetial”, “actibacterial”.
Some abbreviations are not defined or not defined at first use in the text. For instances, “SAR”, “EC50”, etc.
Unless I’m mistaken, there is no compound numbered 2 and 15 and no reason for this is given.
The numbering of Figures should be revised. For instance, Figure S3 is cited in the text before Figures S1 and S2.
Representation of data is not easy to read in every case. Please evaluate the possibility to enhance the readability of Figures.
Figure 1 should be improved, to remove typo errors and enhance its overall presentation. Furthermore, this Figure provides both the rationale of the evaluation plan and some general findings of this work, which can be confusing.
Figure 3 looks neither very useful (redundant with Figures S5-S12) nor easy to read. Table 2 provides main results and should be sufficient. In Figure 3, it is unclear why results for some compounds are not shown (e.g. 4, 6, 8, 10, and 16 with CCUG67180) whereas corresponding data can be found in Table 2.
In Figure 4, strains are noted GU#; standard name of ECOR strains should be used instead or added.
As for the mechanism of action, authors may include the possibility that the antibacterial activity of compounds results from multiple disturbances at various sites of the bacterial cells. Difference in activity may result from difference in terms of interaction with / internalization into target bacteria.
As for perspectives, authors should state about the possible risk of resistance development and persister cells as noted in other studies.
The reasons for conducting the experiments in 2.4 using the dose chosen are not given
Compounds are sometimes referred in the text as e.g. 1, compound 1, or compound (1). Please homogenize and use the format recommended by the Journal.
Line 340-345, please check the sentence.
Line 314, “good selectivity indices” sounds too approximate, please revise/specify.
Structure activity relationships may be discussed in more details. An adaptation of Figure 1 (i.e. an additional Figure) may be provided in Discussion to summarize the SAR unveiled, giving an overall picture of the results obtained in this study.
Authors may also consider/add the following recent publication:
Cantillon D, Goff A, Taylor S, Salehi E, Fidler K, Stoneham S, Waddell SJ. Searching for new therapeutic options for the uncommon pathogen Mycobacterium chimaera: an open drug discovery approach. Lancet Microbe. 2022 May;3(5):e382-e391. doi: 10.1016/S2666-5247(21)00326-8. Epub 2022 Apr 1. PMID: 35544099; PMCID: PMC9042791.
Author Response
Should be published in Antibiotics after minor revision
In their manuscript entitled “Development of dicationic bisguanidine-arylfuran derivatives as potent agents against Gram-negative bacteria”, Catarina Bourgard et al. report the design and synthesis of a series of dicationic derivatives. These compounds were used to examine their antibacterial activity against antibiotics-resistant Gram-negative pathogens of the ESKAPE group and E. coli, cytotoxicity against two human cell lines, and possible mechanism of action.
General appreciation
In my opinion, this study is worth of interest, reporting promising new dicationic bisguanidine-arylfuran derivatives and their thorough characterizations considering a panel of relevant bacterial pathogens. Adequate well-detailed experimental procedures were employed, which allowed obtaining many original data reported in a comprehensive, interesting to read manuscript. I have a list of some only minor suggestions/comments, which is given below. In particular, I think some explanations are missing in multiple instances and data presentation could be improved. Accordingly, I recommend a minor revision before publication of this manuscript in Antibiotics.
Specific comments
- Purity of compounds should be discussed in the text.
Response: NMR spectra along with HRMS analyses establish the identity and show that all final compounds have a higher degree of purity. We have added the following text to the end of the Results/Chemistry section: “Finally, the NMR analyses together with the HRMS studies confirm that all the synthesized compounds have a high degree of purity”.
- More details about the cell lines, their origin and the reasons for their use to determine cytotoxicity, should be given.
Response: A paragraph describing the cell lines and their use in cytotoxicity measurements has been added in the corresponding section of Materials and Methods.
- Authors state about selectivity or therapeutic index. The calculation used is not explicitly given.
Response: We did give a definition of selectivity index in the Table 3 legend. However, we agree that it is a good idea to make this more visible, so the definition is now also stated in Materials and Methods (section on cytotoxicity). We now use “selectivity index” throughout instead of “therapeutic index”, to be consistent.
- The manuscript contains some typo errors. A thorough reading should allow to correct these. For example in Figure 1: “against”, “Essetial”, “actibacterial”.
Response: Figure 1 has been modified, and the typing errors were corrected.
- Some abbreviations are not defined or not defined at first use in the text. For instances, “SAR”, “EC50”, etc.
Response: SAR and EC50 have now been defined at their first use.
- Unless I’m mistaken, there is no compound numbered 2 and 15 and no reason for this is given
Response: Compounds 2 and 15 are intermediates and are shown in Scheme S4. The protocols for their synthesis can be found in Material and Methods, section General Procedure G.
- The numbering of Figures should be revised. For instance, Figure S3 is cited in the text before Figures S1 and S2.
Response: We have renumbered Supplementary Figures according to order of appearance in the text.
- Representation of data is not easy to read in every case. Please evaluate the possibility to enhance the readability of Figures
Response: We have improved visibility in Figures 1,2, and 3, as well as in Table 1, by enlarging structural formulas (Figs. 1, 2 and Table 1) and increasing size of panels (Fig. 3).
- Figure 1 should be improved, to remove typo errors and enhance its overall presentation. Furthermore, this Figure provides both the rationale of the evaluation plan and some general findings of this work, which can be confusing.
Response: Figure 1 has been modified, and now it is clearer according to the proposal of the synthesized compounds and the type errors were corrected.
- Figure 3 looks neither very useful (redundant with Figures S5-S12) nor easy to read. Table 2 provides main results and should be sufficient. In Figure 3, it is unclear why results for some compounds are not shown (e.g. 4, 6, 8, 10, and 16 with CCUG67180) whereas corresponding data can be found in Table 2.
Response: This figure shows in a concentrated form central findings of this paper. We show here side by side the responses from two different human isolates of each bacterial species; one more and one less sensitive to established antibiotics. We highlight which new compounds have a higher selectivity index (blue curves) than the lead compound (red curves) for the respective bacterial strain. For clarity, we show only the most active compounds for each species. So, we think this figure has an important function in our paper.
- In Figure 4, strains are noted GU#; standard name of ECOR strains should be used instead or added.
Response: A new version of Fig 4 with the standard ECOR strain names has now been inserted.
- As for the mechanism of action, authors may include the possibility that the antibacterial activity of compounds results from multiple disturbances at various sites of the bacterial cells. Difference in activity may result from difference in terms of interaction with / internalization into target bacteria.
Response: We have added two sentences mentioning these considerations at the very end of Discussion.
- As for perspectives, authors should state about the possible risk of resistance development and persister cells as noted in other studies.
Response: We have added a short paragraph on these issues in the Conclusion section.
- The reasons for conducting the experiments in 2.4 using the dose chosen are not given
Response: The purpose is to use compound concentrations that partially but not completely inhibit growth on agar. Ideally, all strains from the most sensitive to the most resistant will grow at different rates, such that their growth rate and yield can be quantitated and compared relative to each other. We have inserted a sentence in the Microbial phenomics section of Materials and Methods to this effect.
- Compounds are sometimes referred in the text as e.g. 1, compound 1, or compound (1). Please homogenize and use the format recommended by the Journal.
Response: We have not found an explicit recommendation from this journal. From articles previously published in Antibiotics, we see examples of all three formats (in some cases even mixed in the same paper). We have now changed to use the first style (e.g., 1) throughout the manuscript including Supplementary Information.
- Line 340-345, please check the sentence.
Response: Here, there was a citation formatting error. We were referencing a bioassay screening result for Trypanosoma cruzii reported in PubChem. The citation has now been properly formatted, and the sentence reads fine.
- Line 314, “good selectivity indices” sounds too approximate, please revise/specify.
Response: We have now specified the numbers in the text; selectivity indices from 11 to 24.
- Structure activity relationships may be discussed in more details. An adaptation of Figure 1 (i.e. an additional Figure) may be provided in Discussion to summarize the SAR unveiled, giving an overall picture of the results obtained in this study.
Response: Figure 1 has been modified to better reflect the design and synthesis of the target compounds. We feel that the modified Figure 1 together with discussion gives a good description of the structure activity relationship (SAR).
- Authors may also consider/add the following recent publication: Cantillon D, Goff A, Taylor S, Salehi E, Fidler K, Stoneham S, Waddell SJ. Searching for new therapeutic options for the uncommon pathogen Mycobacterium chimaera: an open drug discovery approach. Lancet Microbe. 2022 May;3(5):e382-e391. doi: 10.1016/S2666-5247(21)00326-8. Epub 2022 Apr 1. PMID: 35544099; PMCID: PMC9042791.
Response: This paper reports successful use of the MMV Pathogen Box to screen for antimycobacterial molecules. This is an interesting case, somewhat parallel to ours, and we now cite it in the Introduction.
Reviewer 3 Report
The manuscript described the Development of dicationic bisguanidine-arylfuran derivatives as potent agents against Gram-negative bacteria along with the mechanism of action. The study is interesting and has the potential to be published. There are some issues to be considered and corrected.
1. The rationale provided in Figure 1 is not clear so it is suggested to modify it. It is not according to the designed and synthesized compounds.
2. Author should move the synthetic schemes into the manuscript as it will be convenient for the readers to understand the chemistry.
3. Author need to add salient features of the newly synthesized compounds in the manuscript e.g NMR peaks etc.
4. Melting points of the final compounds should be provided.
5. Please check the NMR data for compound 3 in supporting information.
6. What solvent system was used to purify the final compound?
7. 1H NMR peaks of the compounds should be assigned.
8. Authors should carefully check the figure numbers provided in the supplementary information as they are not according to the text or figure numbers mentioned in the main manuscript. For e.g. Line 162 ‘Figure S5’ should be ‘Figure S4’.
9. In the results of antibacterial activity, authors should mention the positive results first instead of negative results (lesser potent compounds). In other words, It is better to mention the activity of potent compounds first to come up with a better picture of SAR.
10. Line 173, what is the meaning of ‘activity more or less disappeared’, it should be rephrased.
11. Authors claimed that guanidine substitution is responsible for the anti-bacterial activity, but compound 9 having the guanidine group did not perform well. Please explain the reason for the inactivity of this compound.
12. The EC50 of reference drugs should also be mentioned in the manuscript tables to better compare the results.
13. There are some typographical errors e.g. Line 48, remove ‘e.g.’
Figure 1, typo error, ‘essetial group to keep the activity’. Please check the whole manuscript carefully for these mistakes.
Author Response
The manuscript described the Development of dicationic bisguanidine-arylfuran derivatives as potent agents against Gram-negative bacteria along with the mechanism of action. The study is interesting and has the potential to be published. There are some issues to be considered and corrected.
- The rationale provided in Figure 1 is not clear so it is suggested to modify it. It is not according to the designed and synthesized compounds.
Response: Figure 1 has been modified, and now it is clearer according to the proposal of the synthesized compounds and the typing errors were corrected.
- Author should move the synthetic schemes into the manuscript as it will be convenient for the readers to understand the chemistry.
Response: The chemistry section of Materials and Methods is exceedingly long as it is. We prefer to keep the synthetic schemes in the Supporting Information as otherwise the size of the main manuscript would go out of hand.
- Author need to add salient features of the newly synthesized compounds in the manuscript e.g NMR peaks etc.
Response: All the previously undescribed compounds have been characterized with NMR (proton and carbon) together with High Resolution Mass Spectrometry (HRMS). We thereby prove the identity and purity of the compounds. These are key parameters to generate proper SAR data. However, we don’t see how describing salient features of the newly synthesized compounds improve the manuscript. This is not a paper about the physical/chemical properties of the compounds-the manuscript deals with the antibacterial activity of the compounds.
- Melting points of the final compounds should be provided.
Response: Antibiotics has no specific instructions for how compounds should be characterized or how the data should be presented. We have followed common recommendations for many journals (including Molecules, the sister journal of Antibiotics) that characterization of organic compounds should include, as supplementary data, 1H, 13C and/or other key heteronuclear or 2D NMR spectra, together with HRMS or elemental analysis. Melting points are therefore not included as a part of the characterization of the target compounds.
- Please check the NMR data for compound 3 in supporting information.
Response: There is a good match between the structure of 3 and the NMR spectra: two doublets corresponding to the 1,4 aromatic substitution pattern are clearly observed, with both signals corresponding to two hydrogen atoms. The more shielded singlet in the aromatic area belongs to the hydrogens of the furan ring. Since the furan moiety is symmetrical molecule, we only observe half of the hydrogens. Since the spectrum was obtained in deuterated methanol, hydrogen exchange of hydrogens bound to heteroatoms such as nitrogen will occur. Consequently, the hydrogen atoms on the guanidine moieties do not show up in the 1H NMR spectra.
- What solvent system was used to purify the final compound?
Response: In Materials and Methods, section “General experimental information for synthesis and compound characterization”, you will find the following text: “Purification by flash column chromatography was performed on a Selekt (Biotage, U.K.) automated instrument with Sfär KP-amino D or Sfär silica D cartridges (Biotage, U.K.), mobile phase consist of pentane (solvent A) and ethyl acetate (solvent B). The final compounds were purified by reverse phase flash column chromatography performed on an Isolera (Biotage, U.K.) automated instrument with Sfär C18 D cartridges (Biotage, U.K.), mobile phase consist of water (solvent A) and acetonitrile (solvent B). The standard gradient consisted of x % solvent B for 1 columns volume, x % to y % B for 10 column volumes, and then y % B for 2 column volumes. x and y are defined in the characterization section of the compound the interest.”
- 1H NMR peaks of the compounds should be assigned.
Response: Antibiotics has no specific instructions for how compounds should be characterized or how the data should be presented. Furthermore, many journals state they only want listing of the NMR signals in the experimental part. Unless the journal sees this as an absolute requirement, we do not see the necessity of assigning the NMR signals in the experimental section.
- Authors should carefully check the figure numbers provided in the supplementary information as they are not according to the text or figure numbers mentioned in the main manuscript. For e.g. Line 162 ‘Figure S5’ should be ‘Figure S4’.
Response: “Figure S5” has been changed to “Figure S4” in this position. In addition, we have checked that the supplementary figures are numbered in order of appearance in the text.
- In the results of antibacterial activity, authors should mention the positive results first instead of negative results (lesser potent compounds). In other words, It is better to mention the activity of potent compounds first to come up with a better picture of SAR.
Response: As the relative activity of the compounds against different bacterial species varies, it is difficult to uniformly order them by overall antibacterial activity. Also, much of the improved selectivity of the new compounds stems from reduced cytotoxicity, making it natural to emphasize those aspects first.
- Line 173, what is the meaning of ‘activity more or less disappeared’, it should be rephrased.
Response: This has been rephrased to “activity almost disappeared”.
- Authors claimed that guanidine substitution is responsible for the anti-bacterial activity, but compound 9 having the guanidine group did not perform well. Please explain the reason for the inactivity of this compound.
Response: See Results, section on “Evaluation of cytotoxicity and antimicrobial activity against Gram-negative and Gram-positive non-pathogenic bacterial strains”, third paragraph: “When the phenyl ring in 4 was replaced with a pyridine ring (e.g. in 9) the activity almost disappeared (Tables S1 and S2; Figure S2 B, E, H and K). The loss of activity is attributed to this substitution.
Also see three sentences further down:” The squaryldiamide-based compounds (11-14) were the least active against all bacteria tested, indicating that the guanidino group is essential for antibacterial activity.”
- The EC50 of reference drugs should also be mentioned in the manuscript tables to better compare the results.
Response: We do provide EC50 for the reference drug CTX in Table 2. In Table S3, in addition we give the sensitivity values for amoxicillin, ciprofloxacin, and meropenem.
- There are some typographical errors e.g. Line 48, remove ‘e.g.’
Response: We mean to emphasize that the mechanism of action of guanidine-containing compounds is not established, and that the proposed interaction with the cell surface is just one of several possibilities. We have now reordered the sentence to make this clearer (but kept the “e.g.”)
- Figure 1, typo error, ‘essetial group to keep the activity’. Please check the whole manuscript carefully for these mistakes.
Response: The spelling errors in Figure 1 have been corrected.